# Automated inference of disease mechanisms in patient-hiPSC-derived neuronal networks
Nina Doorn [1] ✉, Michel J. A. M. van Putten [1,2] & Monica Frega [1,3]

Human induced pluripotent stem cells (hiPSCs)-derived neuronal networks on multi-electrode arrays (MEAs) are a powerful tool for studying neurological disorders. The electric activity patterns of these networks differ between healthy and patient-derived neurons, reflecting underlying pathology. However, elucidating these underlying molecular mechanisms requires strenuous additional experiments. Computational models can link observable network activity to underlying mechanisms by estimating biophysical model parameters that simulate the experimental observations, but this is challenging. Here, we address this challenge using simulation-based inference (SBI), a machine-learning approach, to automatically estimate all model parameters that can explain network activity. We show how SBI can accurately estimate parameters that replicate the activity of healthy hiPSC-derived neuronal networks, pinpoint molecular mechanisms affected by pharmacological agents, and identify key disease mechanisms in patient-derived neuronal networks. This demonstrates SBI's potential to automate and enhance the discovery of in vitro disease mechanisms from MEA measurements, advancing research with hiPSC-derived neuronal networks.

Human induced pluripotent stem cells (hiPSCs)-derived neurons have become a key technology to investigate neurological disorders using a patient-specific background in a controlled in vitro environment. Specifically, by differentiating hiPSCs into excitatory neurons through forced expression of *Ngn2*, researchers can rapidly generate electrically mature neuronal networks, whose activity can be easily measured using multi-electrode arrays (MEAs)[1]. In vitro neuronal networks derived from healthy subjects or patients show robust and replicable functional phenotypes on MEAs[2], and various genotype/phenotype correlations have been established using this platform[3–7]. In this way, MEA data from patient-derived neuronal networks can be obtained efficiently with high throughput. Since the electrical activity of the neuronal networks is shaped by underlying molecular mechanisms, the MEA data may contain valuable information about these underlying processes[8]. However, unveiling these mechanisms requires additional extensive and hypothesis-driven in vitro experiments, which are both time-consuming and costly. This is further complicated by the many interacting mechanisms that influence network activity, and the possibility that similar activity patterns can arise from different underlying properties.

Computational models are invaluable in bridging the gap between experimental observations and the (patho-)physiological mechanisms underlying them[9]. Previously, we have developed a biophysically detailed computational model of hiPSC-derived excitatory neuronal networks on MEA that can dissect the effect of specific cellular changes on network activity[10,11]. The parameters of this in silico model describe key physiological characteristics of the neurons, the synapses, and the network connectivity. By adjusting these parameters, simulations that faithfully resemble either healthy- or patient-derived neuronal network activity can be obtained. Differences between the parameters of healthy and disease models reflect altered biological properties in the diseased network.

However, finding the optimal model parameters able to reproduce specific experimental data is difficult. The optimal parameters cannot be calculated explicitly in most biophysical models, because the models are only defined implicitly via stochastic computer simulations. Often, trial-and-error is used to find suitable parameters, but this is a time-consuming and unsystematic approach. An alternative is performing many simulations with different combinations of parameters drawn from a grid or following a specific route through parameter space, and evaluating which simulation matches the experimental observation best[12,13]. These and similar methods have several limitations. First, the user is required to define a distance measure describing how well the simulation replicates the experimental measurement, which can be challenging when observations have many features. Second, these methods quickly become computationally expensive

[1]Department of Clinical Neurophysiology, University of Twente, Enschede, The Netherlands. [2]Department of Neurology and Clinical Neurophysiology, Medisch Spectrum Twente, Enschede, The Netherlands. [3]Department of Informatics, Bioengineering, Robotics and Systems Engineering, University of Genoa, Genoa, Italy. ✉e-mail: n.doorn-1@utwente.nl

when more parameters need to be identified and the amount of required simulations exponentially increases (curse of dimensionality). This is especially difficult because these simulations need to be repeated for every experimental observation (i.e., they are non-amortized). Third, most methods provide a single "best" set of parameters, without information about other possible parameter combinations or the uncertainty of the estimation. This is especially a limiting factor in neuronal modeling where different parameter combinations can result in similar observations (parameter degeneracy)[14], and where activity is naturally robust to some parameter perturbations but very sensitive to changes in other parameters[15].

To address these limitations, simulation-based inference (SBI) was introduced[16–19]. SBI is a machine-learning approach that allows efficient statistical inference of biophysical model parameters using simulations and any prior knowledge or constraints on parameters. SBI can identify not only the best parameters, but the full set of parameter combinations for which the model reproduces the experimental measurements, and their corresponding probabilities (i.e., the posterior distribution). Moreover, the process is amortized, meaning heavy computations only need to be performed once, after which new experimental measurements can be readily analyzed.

Here, we apply SBI to our previously validated computational model of hiPSC-derived neuronal networks on MEA[10]. Our primary aim is to assess the suitability of SBI for the automatic identification of disease mechanisms underlying the phenotype of patient-derived neuronal networks. We obtain the parameters' posterior distribution using multiple experimental MEA measurements from healthy-, patient-derived, and gene-edited neuronal networks. This is complemented by experimental data from neuronal cultures treated with different pharmacological agents. First, we show that the most probable parameters identified by SBI result in simulations with high similarity to experimental measurements. Second, we show that SBI identifies phenotype-causing molecular mechanisms that have previously been proven to be affected in patient-derived networks in vitro. Finally, we provide a set of recommendations for the use of SBI to identify in vitro disease mechanisms from MEA measurements.

## Results

### SBI correctly identifies ground-truth parameters of simulated neuronal-network activity

We used SBI[16] to infer mechanistic information from measurements of hiPSC-derived excitatory neuronal networks on MEA (Fig. 1A). To do so, we sampled 300,000 model parameter configurations from a box prior (Fig. 1A1) with plausible parameter ranges (see "Methods" and Table 1 for details on parameter choices) and used them for simulations with our biophysical computational model (Fig. 1A2). We analyzed the resulting simulations by computing 13 MEA features that capture important characteristics of the network activity and are often used in MEA literature (summary statistics; see "Methods" and Table 2 for details). We then used the analyzed simulations to train a deep neural density estimator (NDE) to identify which model parameter sets produce simulations compatible with experimental measurements (Fig. 1A3). We then evaluated the trained NDE with the MEA features of recordings from healthy and diseased hiPSC-derived neuronal networks (Fig. 1A4, A5). This resulted in a posterior distribution per measurement, which represents the probability of different model parameters given both the prior distribution and the experimental observations (Fig. 1A6). This distribution assigns high probability (yellow) to parameter values that are most consistent with the input measurement, while regions with low probability (dark blue) correspond to parameters that produce simulations that mimic the observation less well (Fig. 1A7).

As an initial evaluation of SBI, we performed parameter inference on simulated data with known model parameters, as exemplified by Goncalves et al.[17]. The resulting posteriors contained the ground-truth model parameters in a high probability region, and the mode of the posterior was close to—or completely overlapped with—the ground truth (Fig. 1B, Supplementary Fig. 1A, B). This illustrates that SBI can correctly identify the parameters of our biophysical computational model. Simulations using the mode closely resembled the input data on which the inference had been

carried out, while model parameters with low posterior probability generated simulations that were distinctly different from the data (Fig. 1C, D, Supplementary Fig. 1C).

### SBI can identify the entire landscape of parameters consistent with MEA measurements of healthy hiPSC-derived neuronal networks

Next, we used SBI to estimate the in silico model parameters that allow us to simulate network activity of healthy in vitro neuronal networks on MEA. For this, we used 10-min measurements of spontaneous activity from neuronal networks derived from a healthy individual (C1 in Frega et al.[3]). Simulations with the mode of the posterior distribution were highly similar to experimental measurements, with no significant differences between most MEA features (Fig. 2A, Supplementary Fig. 2A). To evaluate reproducibility, we also analyzed measurements from another MEA batch derived from the same individual. Interestingly, posterior distributions looked markedly different for measurements of networks from different MEA batches or with different astrocyte batches (e.g., Fig. 2B vs. Supplementary Fig. 2B). This suggests that posterior distributions should only be compared when the experimental measurements originate from the same MEA experiment, in line with previous findings[2].

We noticed that the estimated model parameter distributions, as well as the pairwise marginals, were rather wide, indicating that a large range of different parameters can reproduce the experimental data (Fig. 2B). This could indicate one of two possibilities: either changes in some parameters barely affect the network activity, or changes in several parameters do influence network activity, but their effects compensate for each other[17]. To investigate this, we first assessed how sensitive the model is to variations in different parameters by performing a sensitivity analysis on the posterior distribution. We found that although the model is more sensitive to some parameters than others, no sensitivity scores were negligible (Supplementary Fig. 2D). Thus, we next investigated whether the wide range of possible parameter combinations could be due to parameters compensating for each other. By examining the pairwise marginals, it seems that parameters are only weakly associated, as indicated by large clouds without a clear direction (Fig. 2B). However, pairwise marginals are averages over many network configurations, where all other model parameters may have various values[17]. Even if certain model parameters can vary widely, each value may require a specific configuration of the remaining model parameters. To explore this further, we held all but one or two parameters constant at values sampled from the posterior distribution and observed what values the remaining one or two model parameters could take to still reproduce the desired network activity. This resulted in conditional posterior distributions (Fig. 2C, Supplementary Fig. 2C). We found that these conditional distributions were significantly narrower, confirming that if some model parameters have a certain value, the remaining model parameters can only vary within a limited range—and strongly constrain each other—to still yield simulations that match the experimental observation.

When looking at remaining possibilities with all but two parameters set to a constant value (Fig. 2C), we observed high associations between some model parameters. When generating the conditional distribution over many possible sets, we noticed that some conditional correlations were preserved, leading to an average correlation coefficient significantly different from zero (Fig. 2D, E). We found, for example, a strong negative correlation between AMPA conductance and the probability of connection (Conn%), as well as between NMDA conductance and Conn%. This indicates that fewer synapses can be compensated by stronger synapses and vice versa. Less intuitively, we found a strong positive correlation between the NMDA conductance and the strength of short-term synaptic depression (U (STD)). Literature suggests that both mechanisms strongly but inversely influence the NB duration (NBD) MEA feature[11,20]. To investigate this, we computed the sensitivity of the NBD MEA feature to the model parameters and indeed found NBD to be highly sensitive to both NMDA conductance and the strength of STD (Supplementary Fig. 2E), suggesting these parameters can compensate for each other. To summarize, we show that SBI can identify

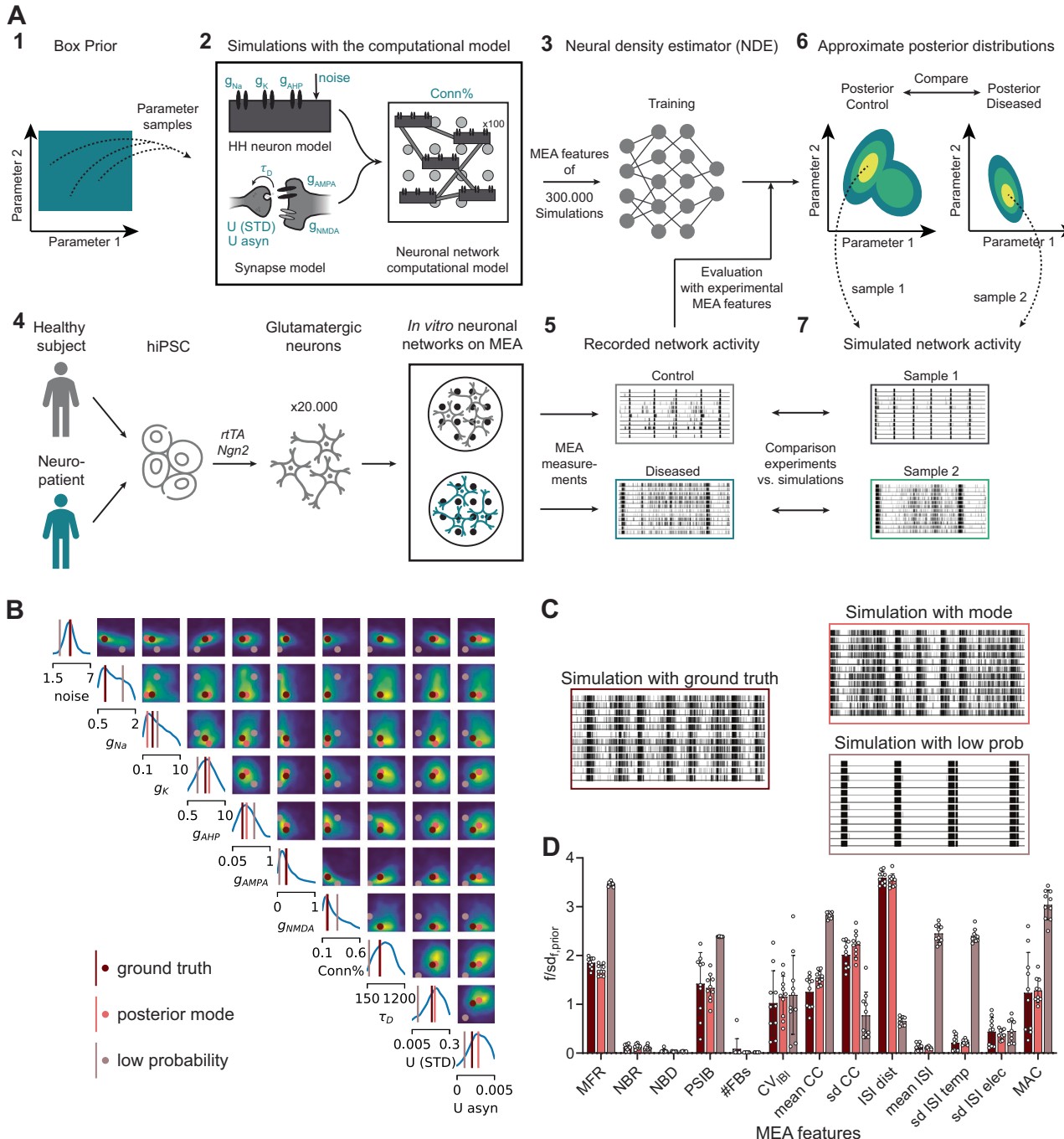

**Fig. 1 | SBI to infer the entire space of parameters consistent with experimental MEA measurements. A** Schematic overview of the method: **1** 300,000 parameter configurations are drawn from within plausible ranges (see Table 1); **2** simulations are performed using our biophysical computational model consisting of 100 Hodgkin-Huxley (HH)-type neurons and AMPAr, and NMDAr-mediated synapses. Model parameters are depicted in blue and reported in Table 1; **3** MEA features (reported in Table 2) of the resulting 300,000 simulations are used to train a neural density estimator (NDE); **4** schematic overview of the protocol to differentiate hiPSCs into neuronal networks on multi-electrode arrays (MEAs): hiPSCs were obtained by reprogramming somatic cells of healthy subjects and patients, excitatory neurons were generated through doxycycline (Dox) inducible overexpression of Neurogenin2 (Ngn2), activity was recorded around DIV 35 with MEA; **5** MEA features are computed from the recorded network activity and used to evaluate the trained NDE; **6** this results in one posterior distribution for each experimental observation, which are

compared to identify mechanistic differences; **7** simulations with parameters sampled from the posterior distribution are similar when compared to experimental observations. **B** Inferred posterior of the ten model parameters given 13 MEA features of simulated activity. Diagonal shows univariate marginals and off-diagonal shows pairwise marginals between the two adjacent parameters, where a lighter color indicates a higher probability. Ground-truth model parameters are shown in brown, posterior mode in pink, and low-probability model parameters in beige. **C** Raster plots showing 1 min of (left) simulation used as input for the inference, (top right) example simulation with the mode of the posterior distribution, and (bottom right) example simulation with the low probability model parameters. **D** MEA features of simulations (n = 10 per condition) with the ground-truth model parameters (brown), the mode of the posterior (pink), and low-probability model parameters (beige). The MEA features are described in Table 2. Data shows mean ± sd. Each feature is normalized by $sd_{f,prior}$, the standard deviation of that feature of simulations sampled from the prior.

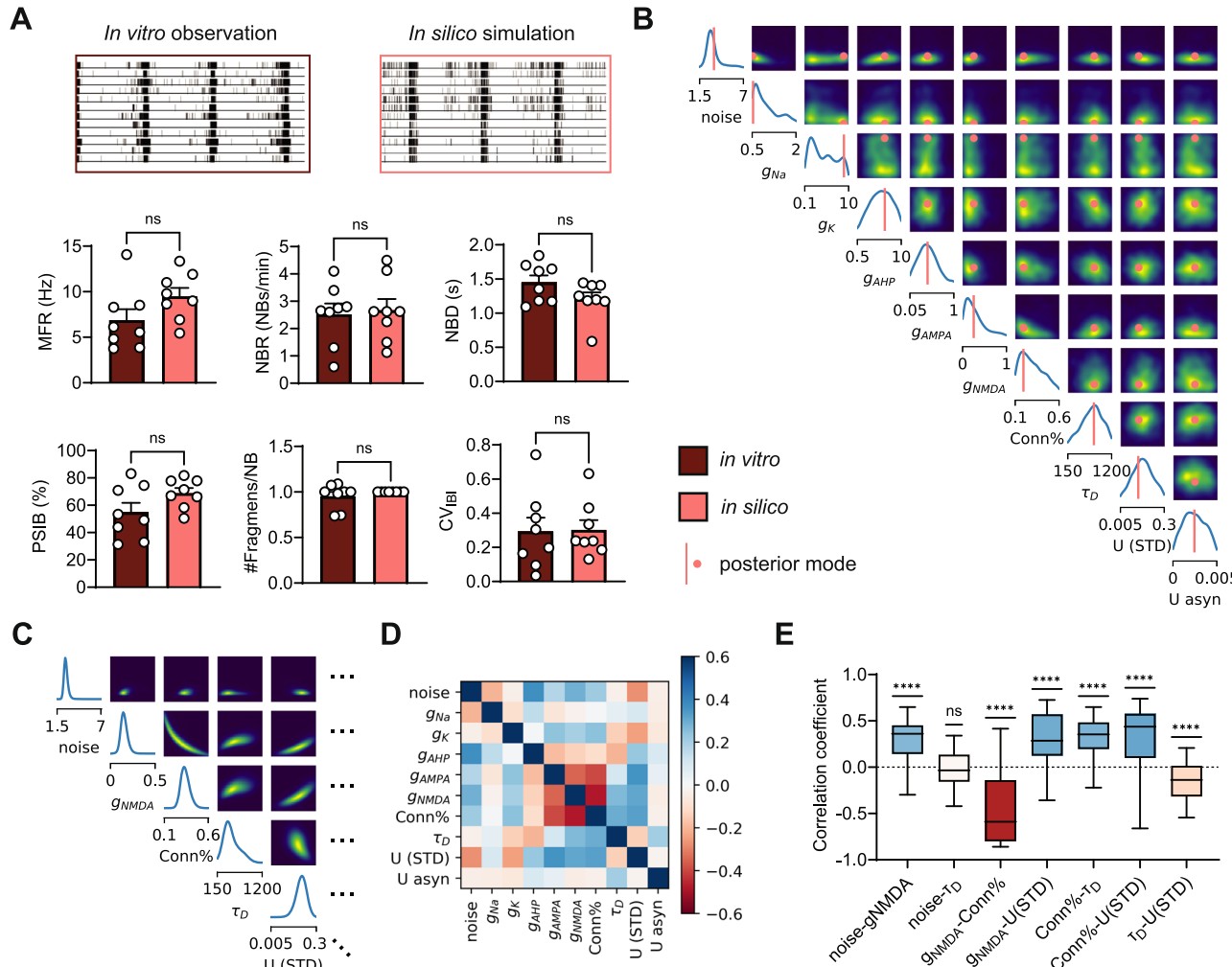

**Fig. 2 | SBI identifies finely-tuned parameters able to generate simulations with high similarity to hiPSC-derived neuronal network measurements.**
**A** Representative raster plot (top left) of spontaneous in vitro neuronal network activity on MEA derived from a healthy individual, used to obtain the posterior distribution, and representative raster plot (top right) from an in silico simulation with parameters from the mode of the inferred posterior distribution. Bottom shows the similarity between features of in vitro measurements ($n = 8$) and in silico mode simulations ($n = 8$). The features are: the network burst (NB) rate (NBR), NB duration (NBD), percentage of spikes in NBs (PSIB), the number of fragments per NB (#FBs), and the coefficient of variation of the inter-burst-intervals ($CV_{IBI}$). Bars show mean $\pm$ SEM. Data are compared with a Mann-Whitney test where ns means $p > 0.05$. **B** Inferred posterior distribution. **C** A subset of a conditional distribution with a particular parameter set. Plots on the diagonal show conditional distribution when all other parameters are fixed. For off-diagonal plots, we keep all but two parameters fixed. Distributions are much more narrow compared to the posterior shown in (**B**). Note the ranges on the x-axis differ. **D** Conditional correlation matrix averaged over 50 conditional distributions. **E** Boxplots of correlation coefficients across 50 conditionals. Colors correspond to average values in (**D**). A one-sample $t$-test was performed to test whether correlation coefficients significantly differed from zero. ns $p > 0.05$, **** $p < 0.0001$.

model parameters able to replicate neuronal network activity on MEA and that these parameters require fine-tuning with respect to each other, even in the presence of wide univariate marginals.

## SBI can identify pharmacological targets in in vitro neuronal networks

To assess whether SBI could identify relevant changes in the physiological properties of hiPSC-derived neuronal networks, we applied it to activity measurements performed on healthy neuronal networks before and after the addition of drugs affecting specific molecular mechanisms. Using SBI, we inferred the posterior distribution for every network before and after the addition of the drug. We compared these distributions per network to see if the parameter differences represented the affected molecular mechanisms.

First, we applied SBI to MEA measurements before and after the addition of Dynasore, which inhibits synaptic vesicle recycling, thereby enhancing the amount of STD[21,22]. In our in silico model, the U (STD) parameter governs the amount of neurotransmitters depleted from the readily releasable pool

upon every pre-synaptic spike, influencing synaptic depression. The $\tau_D$ parameter governs the rate of synaptic vesicle recycling: a higher value indicates slower recycling and thus a longer depression of the synapse. We would therefore expect that Dynasore mainly increases $\tau_D$. When inspecting an example posterior distribution before and after Dynasore addition, it becomes apparent that multiple parameter distributions and their pairwise marginals differ (Fig. 3A). Simulations with the mode of each posterior distributions appear highly similar to the corresponding experimental observations, and the effect of Dynasore on MEA features is very similar in simulations and in vitro (Fig. 3B). Specifically, the NBD was substantially reduced in both cases. We then compared the univariate marginals before and after the addition of Dynasore per network. For one neuronal network (Fig. 3C), we saw a significant shift towards lower values in the marginal of the noise and connectivity, and towards higher values in the marginal of $\tau_D$ and U. To investigate whether these parameter changes were consistent among many networks, we evaluated the significance of these changes for all networks (Fig. 3D). We noticed that while the shift in connectivity was large in some networks but not in all, the shift in the $\tau_D$ distribution was highly significant in

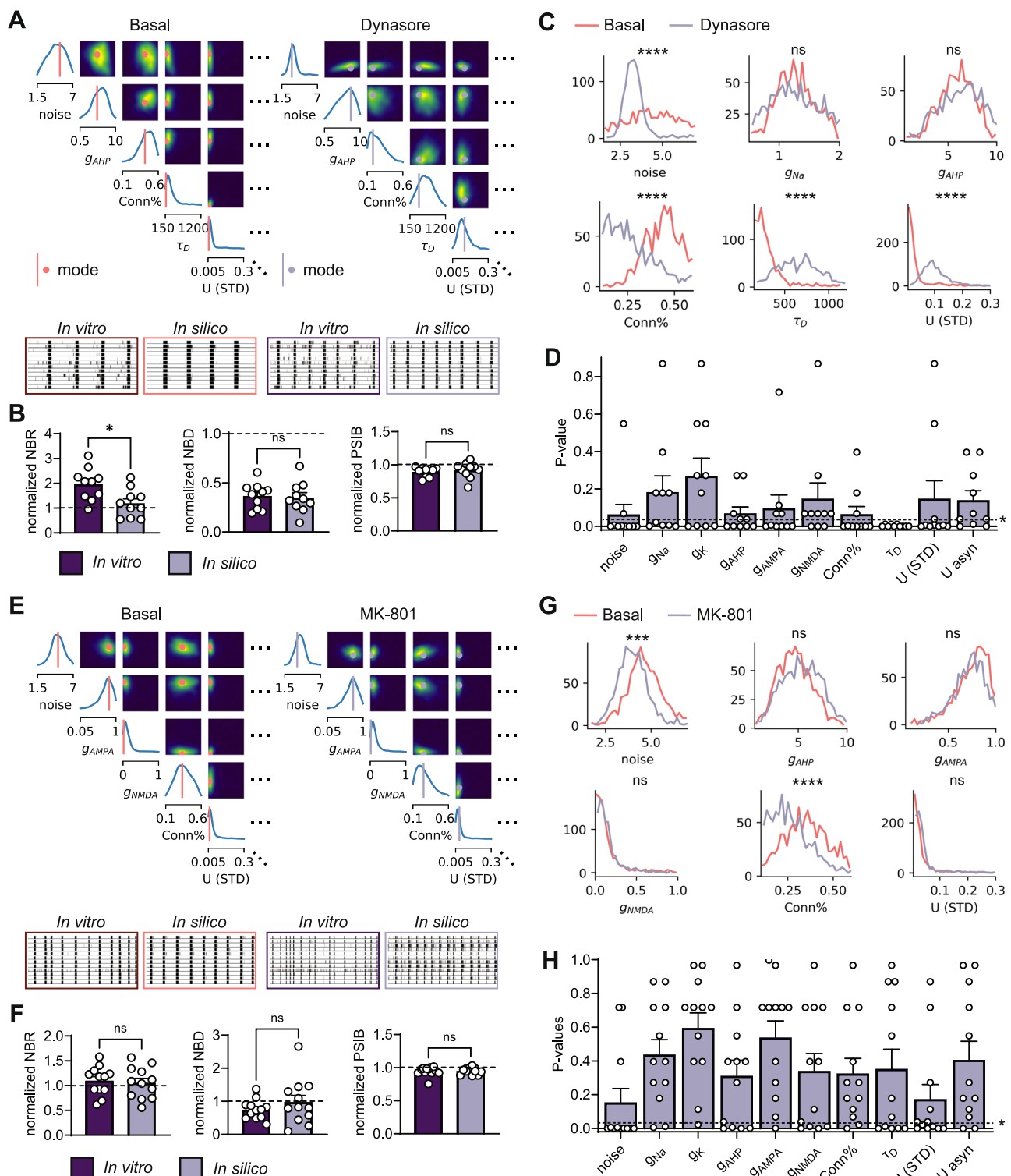

**Fig. 3 | SBI can identify pharmacological targets when network activity is sufficiently altered by drugs. A** An example subset of the posterior distributions inferred from a network activity measurement before (left) and after (right) addition of Dynasore (10 μM), and 1-min raster plots from the in vitro activity and simulated activity with the mode of the posteriors. **B** Normalized features of experiments (n = 10) and simulations (n = 10) before and after adding Dynasore. The features are: mean firing rate (MFR), network burst (NB) rate (NBR), NB duration (NBD), and percentage of spikes in NBs (PSIB). Bars show mean ± SEM. Mann-Whitney test was performed between groups. ns $p > 0.05$, * $0.05 > p > 0.01$. **C** Univariate marginals from the posteriors shown in (**A**). Marginals were compared using a Kolmogorov-Smirnov (KS) test with 50 samples per marginal. ns $p > 0.05$, **** $p < 0.0001$. **D**

$p$ values of the KS-test result of 10 network comparisons per parameter. Bars show mean ± SEM. Dashed line indicates $p = 0.05$. **E** An example subset of the posterior distributions inferred from a network activity measurement before (left) and after (right) addition of MK-801 (1 μM), and 1-min raster plots from the in vitro activity and simulated activity with the mode of the posteriors. **F** Normalized features of experiments (n = 12) and simulations (n = 12) before and after adding MK-801. Mann-Whitney test was performed between groups. ns $p > 0.05$. **G** Univariate marginals from the posteriors shown in (**E**). Marginals were compared using a Kolmogorov-Smirnov (KS) test with 50 samples per marginal. ns $p > 0.05$, *** $p < 0.001$, **** $p < 0.0001$. **H** $p$ values of the KS-test result of 12 network comparisons per parameter.

all networks (Fig. 3D). This demonstrates the importance of comparing multiple networks to eliminate the parameter changes occurring by chance or other factors besides the affected molecular mechanisms.

Next, we applied SBI to MEA measurements before and after the addition of MK-801, an NMDA-receptor blocker[23]. We would expect that MK-801 mainly decreases $g_{NMDA}$. However, the posterior distribution before and after the addition of MK-801 revealed minor differences (Fig. 3E). Simulations with the mode of the posterior distributions resemble in vitro measurements, but the differences between before and after the addition of MK-801 are minimal in both cases (Fig. 3F). In some examples, the distributions of the noise and connection parameters significantly shift towards lower values (Fig. 3G), but these changes were not consistent across networks (Fig. 3H). This suggests that SBI cannot identify molecular changes that do not significantly or consistently affect the neuronal network activity pattern.

### SBI pinpoints known disease mechanisms in patient-derived and genome-edited neuronal networks

To examine whether SBI could identify disease mechanisms in patient-derived neuronal networks, we compiled data from previously investigated and published cell lines. We computed posterior distributions from healthy control networks and compared them to the distributions of patient-derived or genome-edited networks. Because we previously observed that different batches affect SBI results, we only compared posterior distributions of networks cultured and measured on the same MEA plate.

First, we analyzed data from neuronal networks derived from patients with Dravet Syndrome (DS) and Generalized Epilepsy with Febrile Seizures plus (GEFS+), both epileptic encephalopathies linked to a variant in *SCN1A*, which encodes part of the voltage-gated sodium channel $Na_V1.1$ (PAT001 GEFS and PAT001 DS from van Hugte et al.[7]). These networks were cultured alongside networks derived from a healthy individual (Control). Previous in vitro experiments revealed lower sodium current densities, altered action potential (AP) decay dynamics and firing patterns suggestive of a reduction in potassium currents, and reduced spontaneous excitatory post-synaptic current (sEPSC) frequency and amplitudes in GEFS+ and DS networks compared to Control[7,10]. Moreover, increased STD was recently suggested to be important in GEFS+ and DS networks to explain the phenotype[11]. We found that the posterior distributions inferred with SBI visually differed between healthy control networks and DS and GEFS+ (Fig. 4A). Univariate marginals of several parameters also significantly differed in both independent MEA batches (Fig. 4B and Supplementary Fig. 3A). Consistent over two batches, the noise, sodium conductance, and potassium conductance distributions significantly shifted towards lower values in both GEFS+ and DS (Fig. 4C). The strength of STD distribution significantly shifted towards higher values. Additionally, DS-inferred distributions showed a shift towards higher NMDA conductances, and GEFS+-inferred distributions showed a shift towards lower connectivity and higher asynchronous release.

Second, we analyzed networks differentiated from isogenic iPSC lines with a monoallelic frameshift variant in exon 8 of *CACNA1A*, generated via CRISPR/Cas9[24]. *CACNA1A* encodes part of the voltage-gated calcium channel $Ca_V2.1$, crucial for proper neurotransmitter release[25]. These networks were cultured alongside with networks derived from a healthy individual (Control). In vitro experiments revealed a significantly reduced number of synapses, altered AMPA signaling, and reduced potassium channel functioning. Per MEA, we analyzed the shift in univariate marginals between the isogenic networks (Fig. 4E). Averaged over three batches (Fig. 4F, Supplementary Fig. 3B), we found a significant shift towards lower values in the distributions of the noise parameter, the AMPA conductance, and the connectivity. There was a significant negative shift in the potassium conductance in only one batch.

### Discussion

MEAs provide a rapid means to gather neuronal electrophysiological data from patient-derived or genome-edited neuronal networks. These data

often exhibit distinct characteristics that differ between healthy and diseased networks, reflecting underlying pathological molecular mechanisms[2,3,5–7,24]. While some mechanisms are explicitly linked to specific activity patterns[11,20], characterizing them definitively through quantitative analysis of the data remains challenging. Additional experiments to nonetheless uncover these mechanisms, such as patch-clamp measurements or RNA sequencing, are costly and time consuming and may not cover all possible molecular pathways, potentially overlooking crucial mechanisms[10,24]. Here, we propose the use of SBI and our previously developed biophysical computational model to automatically identify disease mechanisms in cultured neuronal networks on MEAs. While SBI shows promise for accurately inferring biological and biophysical parameters[17], its application to uncover pathophysiology in MEA data has not been previously reported.

To investigate whether SBI could pinpoint specific molecular pathways affected in neuronal networks, we applied it to pharmacologically modulated networks. When using Dynasore (an endocytosis inhibitor[21]), SBI accurately predicted an increase in the time constant governing synaptic vesicle endocytosis across all networks. Additionally, in most networks, SBI predicted a reduction in network connectivity and an increase of U, describing the strength of STD. This aligns with the expected outcome, as prolonged synaptic vesicle depletion leads to enhanced synaptic depression and reduced functional connectivity[22]. Using MK-801, an NMDAr-blocker, SBI predicted a decrease in connectivity, but this prediction was not consistent across all networks. Moreover, SBI predicted an initial NMDAr conductance close to zero. This appears in line with the lack of a clear effect of MK-801 on the MEA activity features in most networks. This shows that SBI, as well as other parameter inference techniques, can only identify changes that are already clearly and consistently reflected in the characterization of system behavior used as input. Our MEA features might not fully capture the change in network activity induced by MK-801.

Using cultured neuronal networks from patients with GEFS+ and DS, both harboring a variant in *SCN1A* encoding part of the voltage-gated sodium channel, SBI successfully predicted lower values of the voltage-gated sodium channel conductance for both diseases, with the change more pronounced in DS across the two batches. This is in agreement with voltage clamp measurements in GEFS+ and DS neurons, where DS neurons show more severely impaired sodium currents[7]. SBI also predicted lower values of the voltage-gated potassium channel conductance in both GEFS+ and DS, albeit more prominently in GEFS+ on average. This appears in line with the current clamp measurements demonstrating impaired AP decay dynamics in both conditions, albeit more markedly in GEFS+[7]. Additionally, SBI estimated significantly higher values of STD strength in both GEFS+ and DS, along with a larger STD time constant for DS and increased asynchronous release in GEFS+. These inferences align with previous findings implicating these mechanisms in the observed phenotypes of GEFS+ and DS[11]. Furthermore, lower sEPSC amplitudes and frequencies were observed in DS neurons, possibly indicating more depressed synapses[10].

Next, we investigated CRISPR/Cas9 edited networks with haploinsufficiency of *CACNA1A*, a gene encoding part of the voltage-gated calcium channel. Averaged across three independent batches, SBI predicted a significantly lower connection probability in $CACNA1A^{+/-}$ neuronal networks. This aligns with the immunocytochemistry results showing a significantly reduced number of synapses in $CACNA1A^{+/-}$ networks[24]. Furthermore, SBI suggested a reduction in AMPA conductance. Although direct measurements of the AMPA current were not conducted in vitro, pharmacological testing revealed $CACNA1A^{+/-}$ network activity was governed more by GluA2-lacking AMPA receptors (AMPArs). Our computational model only contains GluA2-containing AMPA receptors with fast opening and closing dynamics and NMDA receptors (NMDArs) with slower opening and closing dynamics and dependence on repetitive firing. GluA2-lacking AMPA receptors resemble the dynamics of the NMDA receptors present in our in silico model[26]. Therefore, a shift towards more influence of GluA2-lacking AMPA receptors in vitro could be predicted by SBI as a reduction in GluA2-containing receptors, leading to a more

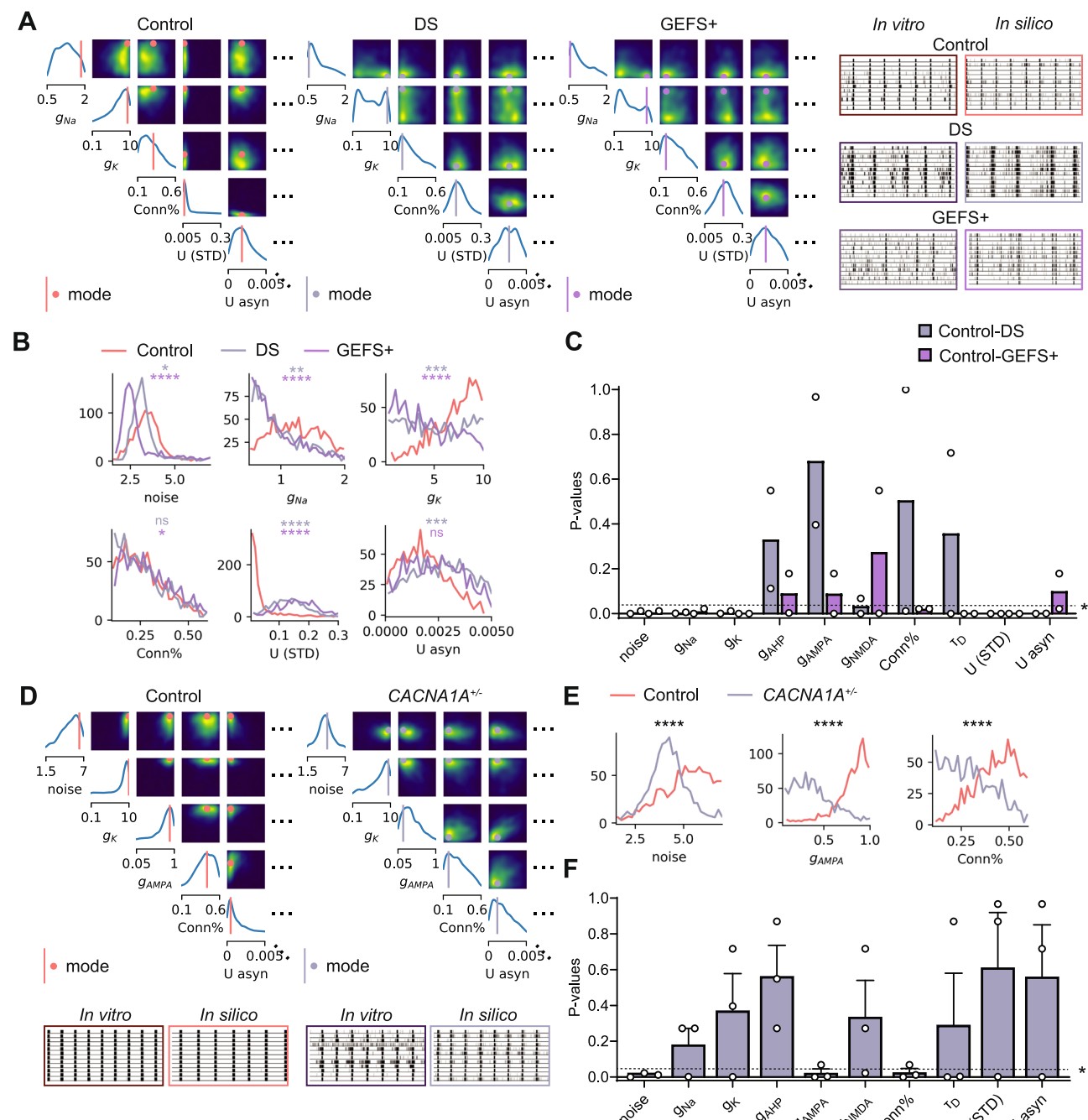

**Fig. 4 | SBI identifies known disease mechanisms in patient-derived and genome-edited neuronal networks. A** Left: Example subsets of posterior distributions inferred from average neuronal network activity of a healthy Control, a patient with Dravet Syndrome (DS), and a patient with Generalized Epilepsy with Febrile Seizures plus (GEFS+). Right: representative 1-min raster plots of spontaneous network activity of the different cell lines and the corresponding simulations with the mode of the posterior distributions. **B** Univariate marginals from the posteriors shown in (**A**). Marginals were compared using a Kolmogorov-Smirnov (KS) test with 50 samples per marginal. ns $p > 0.05$, * $p < 0.05$, ** $p < 0.01$, *** $p < 0.001$, **** $p < 0.0001$. **C** $p$ values of the KS-test result of 2 batches per parameter. Bars show mean. Dashed line indicates $p = 0.05$. **D** Top: Example subsets of posterior distributions inferred from average neuronal network activity of Control and $CACNA1A^{+/-}$ networks. Bottom: representative 1-min raster plots of the spontaneous network activity from the isogenic lines and corresponding simulations with the mode of the posterior distributions. **E** Univariate marginals from the posteriors shown in (**D**). **F** $p$ values of the KS-test result of 3 batches per parameter. Bars show mean ± SEM.

NMDA-dominated synaptic transmission. In $CACNA1A^{+/-}$ neurons, evidence was also found for reduced voltage-gated potassium channel function, indicated by the slightly lower AP decay time, reduced AHP amplitude, reduced rheobase, and altered potassium channel-related RNA expression[24]. SBI predicted a significant shift towards lower values of the potassium-channel conductance. However, this shift was observed in only one batch. The lack of an effect in the other batches might be due to the small size of the

effect measured in vitro, or to the model's relative insensitivity to the potassium channel parameter (Figure S2D).

SBI predicted a lower amplitude of the noisy membrane potential fluctuations in GEFS+, DS, and $CACNA1A^{+/-}$ networks. This noise variable represents both the effect of spontaneous neurotransmitter release on the membrane potential—which is dependent on the number of synapses, their strengths, and their spontaneous release frequency—and the effect of

electrical noise on the membrane potential—which is dependent on the input resistance of the neurons, their size, and the propensity of ion channels to randomly open and close[27]. In DS neurons, we recorded lower sEPSC amplitudes and frequencies, which would explain the decreased effect of these events on the membrane potential fluctuations. In $CACNA1A^{+/-}$ neurons, the synaptic calcium channels are impaired. Spontaneous opening of these channels underlies part of the spontaneous neurotransmitter release, and blocking these channels has been shown to reduce miniature EPSC frequency[28]. Thus, the impaired noise in $CACNA1A^{+/-}$ neurons may be explained by the impaired calcium channels. However, $CACNA1A^{+/-}$ neurons also showed significantly increased soma sizes, which would decrease the input resistance and thus reduce the effect of channel noise on the membrane potential, which may also explain a reduction in the noise parameter[27].

Employing SBI with our biophysical model presents inherent challenges. SBI relies on the assumption that the computational model sufficiently captures the biological processes underlying the experimental observations. If the model does not accurately reflect the system it aims to describe—known as model misspecification—then even a well-calibrated inference procedure can return misleading parameter estimates[29]. If the simulator is incomplete, inferred parameters may compensate for unmodeled effects rather than reflecting true biophysical properties. While our model is previously validated with in vitro experiments[10,11], all models remain simplifications. To assess potential misspecification, we applied the diagnostic test proposed by Schmitt et al.[30] and found no clear evidence of model misspecification for our model and experimental data (Supplementary Fig. 4).

Nevertheless, some disease mechanisms may remain unmodeled. For instance, mitochondrial function is not explicitly included. Moreover, we inferred only 10 model parameters while keeping others fixed, which may limit the approach's flexibility in capturing diverse disease mechanisms and experimental observations. Expanding the number of inferred parameters would increase computational costs due to the need for more simulations to train the NDE. Additionally, it could complicate the interpretability of the inferred posterior distributions, due to parameter redundancy and the widening of marginals. While inferring all model parameters and fixing uninformative ones post hoc could aid interpretation, this was not computationally feasible here.

A key limitation of any in vitro-based approach is the gap between the in vitro system and the complexity of the human brain. While in vitro neuronal networks offer a valuable experimental platform for studying disease-related changes in neuronal function, they are much simplified with respect to the in vivo neuronal environment. Therefore, it is essential to exercise caution when extrapolating findings beyond the in vitro system. While our in silico model accurately reflects the in vitro system, it may not capture all mechanisms relevant to pathology in vivo. For example, the in vitro system lacks inhibitory neurons and patient-derived astrocytes. Although our model is not misspecified in this regard (since it accurately reflects the experimental conditions), mechanisms related to inhibitory or astrocyte dysfunction cannot be inferred. However, if inhibitory neurons and patient-derived astrocytes were to be included in future experiments, the model could be extended accordingly to enable inference of inhibition- or astrocyte-related mechanisms[31,32].

Another limitation of SBI is the need to define prior distributions and summary statistics describing the experimental data, which can introduce biases. We chose a uniform prior as was done in similar research[17,33]. We based prior bounds on biological plausibility as well as experience with model parameter configurations resulting in realistic MEA feature values. SBI can be trained on hand-crafted summary features, as well as raw time series data, or embeddings learned by neural embedding networks, among others. As our goal is not to replicate the full time series, but rather properties that remain consistent across multiple experimental replicates, we chose the use of hand-crafted summary statistics. Yet, to assess whether data-driven summary statistics might improve inference, we also implemented a 2D-convolutional embedding network trained jointly with the NDE. However,

this approach resulted in washed-out posteriors, poorer parameter recovery, and a dramatically increased computational cost of inference. Full details are provided in the Supplementary Methods and Supplementary Fig. 5. Instead, we opted for predefined MEA features due to their robustness across different recording durations and settings, their widespread use in the MEA field, and their interpretability in relation to known (patho)physiological mechanisms[2,3,7,34]. This facilitates comparisons across studies and ensures relevance within the MEA research community. A subset of these MEA features depend on the proper detection of NBs, which can be difficult in the largely varying set of simulations[35]. To limit biases, NB detection and MEA feature calculation have been performed identically in the in vitro measurements and in silico simulations. Additionally, we assessed the performance of SBI with NDEs trained on different subsets of MEA features (Supplementary Fig. 6). This showed that while NB-detection dependent features do not harm inference, the addition of the other features improves inference.

A final limitation of SBI is the difficulty in verifying the correctness of the inference. We performed posterior predictive checks to see if the ground-truth parameters were contained within high-probability regions. Recently, more posterior checks have become available that could be employed by future users to further assess the validity of inference[36,37]. However, we lacked the means to validate if the entire posterior was estimated properly. Since the likelihood of our model is intractable, computing a ground-truth posterior distribution for comparison was not feasible. Yet, SBI previously proved to be accurate in estimating the posteriors of several neural dynamics models where likelihood-based parameter estimation was possible, enabling proper validation[17]. Nevertheless, corroborating SBI predictions with targeted in vitro experiments is imperative when investigating disease mechanisms.

Despite these limitations, we demonstrate that SBI offers several advantages over traditional parameter estimation methods. While trial-and-error methods are common in neuronal modeling studies, they lack systematicity and often yield singular solutions[10,38,39]. Similarly, parameter-searching methods, such as grid searches or evolutionary algorithms, require defining distance measures to experimental observations, necessitate numerous simulations for every experimental observation, and provide only one optimal parameter set without quantifying uncertainty or parameter importance[12,13]. Furthermore, numerous studies highlight parameter degeneracy in biological neuronal networks and computational models: the same activity can arise from vastly different parameters[14,40]. Recognizing this *nonuniqueness* is crucial to avoid drawing incomplete conclusions when observing changes in parameter values between experimental conditions. SBI addresses these challenges by identifying the full range of parameters that explain the observations, unlike other parameter estimation techniques. Finally, SBI is amortized, meaning that performing the many simulations and training the NDE only needs to be executed once. If new experimental data becomes available, a corresponding posterior distribution can be calculated in a matter of seconds. Therefore, SBI represents the most compelling approach for integrating parameter inference into the standard MEA data analysis pipeline.

Based on our findings, we propose that SBI holds promise for automatically detecting disease mechanisms in patient-derived neuronal networks on MEA. However, several important recommendations and considerations should be taken into account when employing this method. Firstly, it is crucial to compare neuronal networks cultured on the same MEA and measured simultaneously, as differences in astrocyte and MEA batches can introduce confounding factors. Principal-components analysis conducted previously highlighted the influence of these factors on MEA data features[2], and here we observed that they also influence SBI-identified posterior distributions. Secondly, conducting multiple comparisons using independent neuronal preparations on MEA is essential to discern changes consistent across batches. Previous advice emphasized the importance of employing a sufficient number of experimental batches[2], and here we observed that not all parameter differences were reproduced across batches. We thus strongly recommend performing sufficient

experimental replicates and comparing univariate marginals per MEA to identify parameters that are altered consistently. Thirdly, targeted in vitro investigations of SBI-identified disease mechanisms are imperative to validate predictions. In some cases, invalid inferences could result from undetected model misspecification. Additionally, SBI identifies a multitude of parameter possibilities consistent with the data, resulting in multiple predicted parameter changes of which potentially only a subset is present in vitro. Investigating all possibilities suggested by SBI can provide clarity on genuine in vitro changes. Nevertheless, SBI significantly reduces the number of in vitro experiments required to properly identify disease mechanisms. Without SBI, researchers would face the daunting task of conducting endless experiments to explore all possibilities, or risk over-looking crucial mechanisms by basing experiments solely on initial hypotheses. With SBI, a trained NDE, and the analysis pipeline outlined here, MEA activity can be analyzed easily and rapidly to automatically identify all the possible disease mechanisms able to explain patient-derived neuronal network phenotypes. This paves the way for targeted experiments and novel insights into disease.

## Materials and methods
### In vitro MEA data acquisition
We used data from neuronal networks derived from hiPSCs, previously cultured and recorded on MEA. hiPSC cells were differentiated into excitatory cortical Layer 2/3 neurons through doxycycline-inducible over-expression of *Neurogenin 2* (*Ngn2*) as described previously[1,2].

Informed consent was obtained from all subjects before using their hiPSCs, and approval was obtained from the medical ethical committee of Radboud University Medical Center, Nijmegen (2018-4525). All ethical regulations relevant to human research participants were followed.

The control line used for both pharmacological experiments and as control with diseased and gene-edited networks was obtained from the Coriell Institute (GM25256, RRID: CVCL_Y803) and was reprogrammed from skin fibroblasts of a 30-year-old healthy male. It was previously described and characterized[3,41].

Drug manipulations were previously performed on neuronal networks derived from the control line as described in Doorn et al.[11] for Dynasore and in Doorn et al.[10] for MK-801, using the 24-well MEA system (Multichannel systems, MCS GmbH, Reutlingen, Germany), with a sampling frequency of 10 kHz.

MEA recordings of neuronal networks derived from patients with GEFS+ and DS were kindly provided by Nael Nadif Kasri and Eline van Hugte, and are described (Control, PAT001 GEFS, PAT001 DS) in van Hugte et al.[7]. Activity was recorded on the 24-well MEA system (Multi-channel systems, MCS GmbH, Reutlingen, Germany) for 10 min with a sampling frequency of 10 kHz.

MEA recordings of control and *CACNA1A*$^{+/-}$ networks were kindly provided by Nael Nadif Kasri and Marina Hommersom, and are described in Hommersom et al.[24]. Activity was recorded on the Maestro Pro MEA system (Axion BioSystems, Atlanta, GA, USA) for 5 min with a sampling frequency of 12.5 kHz. Because this MEA system had 16 electrodes per well, we omitted the corner four electrodes from analysis, to mimic the electrode topology of the Multichannel MEA system and the computational model.

During all recordings, the temperature was maintained at 37 °C, and a flow of humidified gas (5% CO$_2$ and 95% ambient air) was applied onto the MEA plate.

### Computational model
The in silico computational model is described in Doorn et al.[10], with the addition of asynchronous neurotransmitter release described in Doorn et al.[11]. In short, the model consists of one hundred Hodgkin-Huxley neurons with leaky channels and voltage-gated sodium and potassium channels. We also include sAHP channels, modeled as potassium channels whose conductance increases upon AP firing. The neurons are hetero-geneously excitable through a variable constant external input current,

**Table 1 | Overview of the parameter value ranges of the free parameters, and values of the fixed parameters of the computational model**

| Parameter | Description | Value or Range | Unit |
|---|---|---|---|
| noise | Standard deviation of the noisy fluctuations | [1.5–7] | mV |
| $\bar{g}_K$ | Maximum delayed rectifier potassium conductance | $[0.1, 10] \cdot 5$ | mS · cm$^{-2}$ |
| $\bar{g}_{Na}$ | Maximum voltage-gated sodium conductance | $[0.5, 2] \cdot 50$ | mS · cm$^{-2}$ |
| $g_{AHP}$ | Maximum AHP conductance | [0.5, 10] | nS |
| $\bar{g}_{AMPA}$ | Maximal conductance of AMPA | [0.05, 1] | nS |
| $\bar{g}_{NMDA}$ | Maximal conductance of NMDA | [0, 1] | nS |
| Conn% | Probability of two neurons connecting | [0.1, 0.6] | - |
| $\tau_D$ | Recovery timescale of STD | [150, 1200] | ms |
| U (STD) | Strength of STD | [0.005, 0.3] | - |
| U asyn | Strength of asynchronous release | [0, 0.005] | - |
| area | Neuron Size | 300 | µm$^2$ |
| $C_m$ | Membrane capacitance | 1 | µF · cm$^{-2}$ |
| $\bar{g}_l$ | Leak conductance | 0.3 | mS · cm$^{-2}$ |
| $E_K$ | Nernst potential of potassium | −80 | mV |
| $E_{Na}$ | Nernst potential of sodium | 70 | mV |
| $E_l$ | Nernst potential of the leak current | −39.2 | mV |
| $V_T$ | Potential to adapt spike threshold | −30.4 | mV |
| $\alpha_{Ca}$ | Strength of sAHP | 0.00035 | - |
| $\tau_{AHP}$ | Recovery timescale of sAHP-currents | 6 | s |
| $E_{AMPA}$ | Nernst potential of AMPA | 0 | mV |
| $E_{NMDA}$ | Nernst potential of NMDA | 0 | mV |
| $\alpha_{NMDA}$ | Multiplicative constant of NMDA dynamics | 0.5 | kHz |
| $\tau_{AMPA}$ | Decay time for AMPA synapses | 2 | ms |
| $\tau_{NMDA,decay}$ | Decay time for NMDA synapses | 100 | ms |
| $\tau_{NMDA,rise}$ | Rise time for NMDA synapses | 2 | ms |
| Maxdelay | Maximum conduction delay | 25 | ms |
| sd$_w$ | Standard deviation of synaptic weights distribution | 0.7 | - |
| $\tau_{Asyn}$ | Recovery timescale of asynchronous release | 700 | ms |
| $U_{max}$ | Saturation level of asynchronous release | 0.5 | 1/ms |
| $x_0$ | Amount of neurotransmitter in a vesicle | 5 | - |

accounting for intrinsic differences. In addition, we induce stochastic fluctuations of the membrane potential of neurons to mimic the effect of spontaneous neurotransmitter release or channel noise. Neurons are connected to a fraction of other neurons through synapses. These synapses include models of AMPARs, which open immediately upon arrival of a pre-synaptic spike and close rapidly. Additionally, we include NMDARs, which open and close slowly. The NMDARs are blocked by magnesium ions that are removed upon depolarization of the post-synaptic neuron. The strengths of the synapses vary slightly and are modulated by STD, following the Markram-Tsodyks model[42].

Pre-synaptically, neurotransmitters are released synchronously (time-locked to the arrival of the AP), and asynchronously (where the release probability increases upon repetitive firing), following the Markram-

Tsodyks model extension by Wang et al.[43]. Neurons were placed on a grid, allowing for the inclusion of distance-dependent conduction delays and virtual electrodes, mimicking MEA measurements.

To ensure computational feasibility and model interpretability, we fixed specific model parameters related to membrane and synaptic time constants while allowing others, such as ion channel conductances and synaptic strengths, to vary. The membrane time constant emerges from passive properties such as the membrane capacitance and leak conductance, both of which remain stable in most disease conditions[44]. AMPAr and NMDAr decay are fixed because they are determined by receptor subunit composition, which is developmentally regulated and remains stable under most pathological conditions[45]. Ion channel time constants are highly correlated with the expression of that ion channel[14], so allowing both time constants and maximal conductances to vary would result in parameter redundancy and less interpretable posteriors. Therefore, we fixed time constants while keeping ion channel conductances free to vary. Lastly, Nernst potentials were fixed because they are mainly governed by astrocyte function. Since our cultures have healthy astrocytes, not patient-specific ones, we assume the astrocyte-related contributions to remain constant and unrelated to disease mechanisms. Fixed parameter values and prior ranges of the free parameters can be found in Table 1.

Forward simulations were performed using the Brian2 simulator[46]. One simulation took about two to three times the simulated time on a regular desktop CPU. To parallelize simulations across multiple desktop computers, we used a distributed computing method with dynamical load balancing[47]. While this platform significantly accelerates computation and facilitates flexible resource usage, it is not essential to the simulations themselves—similar performance could, in principle, be achieved with alternative parallel computing architectures.

### Pre-processing and feature extraction

In vitro MEA recordings and in silico virtual electrode recordings are pre-processed and analyzed identically. Signals were filtered between 100 and 3500 Hz using a fifth-order Butterworth filter. We detected spikes using an amplitude threshold-based method, where the threshold was four times the root mean square of the electrode signal.

The computed MEA features are summarized in Table 2. The network firing rate was computed by binning spikes at all electrodes in 25 ms time bins. To detect NBs, we employed two thresholds on this network firing rate to start and stop the NB, set to 1/4th and 1/50th of the maximum firing rate

respectively. Additionally, 50% of the active electrodes (i.e., electrodes with an average firing rate above 0.02 Hz) had to be firing during the NB, and the firing rate should remain above the NB start- or stop-threshold for 50 ms to start or end NB detection. To detect NB Fragments, we smoothed the network firing rate by convolution with a Gaussian kernel and used a peak detection algorithm on the smoothed network firing rate where peaks should have a minimal height of 1/16th of the maximum firing rate and a minimal prominence of 1/20th of the maximum firing rate. From this NB detection, we constructed five features. Specifically, we calculated the NB Rate (NBR), describing the number of NBs per minute, the NBD, describing the time between the start and stop of the NB in seconds, the Percentage of Spikes in NBs (PSIB), describing the percentage of spikes contained in the detected NBs, the number of fragments per NB (#FBs), and the coefficient of variation of the inter-NB-intervals ($CV_{IBI}$).

We computed eight additional features describing other characteristics of the data. The first feature was the mean firing rate (MFR). Then we computed the continuous inter-spike-interval (ISI) time-series per MEA electrode, by determining the time between the previous and next detected AP for every time point. From these ISI-time-series, we computed the average ISI (mean ISI) across all timepoints and electrodes. By first taking the average ISI-time-series across electrodes, we defined the standard deviation of the ISI over time (sd ISI temp), and by first taking the average ISI across time points, we defined the standard deviation of the ISI over electrodes (sd ISI elec). Moreover, we computed the sample correlation coefficient between every pair of electrodes ISI-time-series. From this, we computed the average correlation coefficient (mean CC) and the standard deviation of the correlation coefficients (sd CC). Finally, we computed the average ISI-distance between every pair of electrodes as described in Kreuz et al.[48]. The above measures describe the spiking behavior of a network as well as the variation of spiking behavior temporally and spatially. Finally, to have a measure for the strength of oscillations in the network activity without relying on NB detection, we computed the Maximum Auto-correlation Component, as described in Maheswarenathan et al.[34].

We trained the NDE on two additional features, the average and standard deviation of the correlation coefficient between binarized spike trains. However, we observed high correlations (0.96) between these features and the mean CC and sd CC of the ISI-time-series. Consequently, we chose not to show these features in the figures and we argue they could be omitted from training of the NDE.

### Simulation-based-inference

To estimate the posterior distribution of the free parameters of the computational model, we employed amortized single-round neural posterior estimation[17] using the SBI toolbox[16] in a Python 3.9 environment. We sampled 300,000 parameter configurations from the prior distribution (a uniform distribution within the ranges shown in Table 1) and performed 3-min simulations with the computational model with each of these configurations. For every simulation, we computed the 13 MEA features described above. Next, we trained an NDE on these parameter configurations and resulting MEA features. This NDE was the standard Masked Autoregressive Flow provided by the SBI toolbox. Then, we evaluated the NDE with MEA features from either simulated network activity (for posterior-predictive checks) or experimental network activity. This resulted in a posterior distribution for each observation. Posterior distributions were visualized by plotting the marginals of 1000 samples drawn from the distribution.

As a check for the accuracy of the inference, we performed posterior-predictive checks. Initially, we used ground truth model parameters to generate simulations. The trained NDE was then used to estimate the posterior distribution of model parameters consistent with these simulations. The univariate and pairwise marginals were then inspected to see whether the ground truth model parameters fell within regions with a probability above 50% of the maximal probability. We performed posterior-predictive checks with NDEs trained on 100,000, 200,000, and 300,000 simulations, to see when the posterior distributions converged and

### Table 2 | Overview of the MEA features used for training of the NDE

| MEA feature | Description | Unit |
|---|---|---|
| MFR | Mean Firing Rate | Hz |
| NBR | Network Burst Rate | 1/min |
| NBD | Network Burst Duration | s |
| PSIB | Percentage of Spikes in NBs | - |
| #FBs | Average number of fragments per NB | - |
| $CV_{IBI}$ | Coefficient of Variation of the Inter-NB-Intervals | - |
| mean ISI CC | Average Pearson Correlation Coefficient between ISI-timeseries | - |
| sd ISI CC | Standard deviation of the Pearson Correlation Coefficients | - |
| ISI dist | Average ISI distance between ISI-timeseries of electrode-pairs | - |
| mean ISI | The average ISI between all electrodes and timepoints | s |
| sd ISI temp | Standard deviation of ISI-values averaged between electrodes | s |
| sd ISI elec | Standard deviation of ISI-values averaged over time | s |
| MAC | Maximum Autocorrelation Component | - |

checks were sufficient (all twenty-five checks in 50% or higher regions). To visualize the accuracy of the estimation, we performed ten simulations with the mode of the posterior distribution, the ground truth model parameters, and low-probability model parameters, and quantified the MEA features. For visualization, we normalized the MEA features by the standard deviation of that feature in the 300,000 simulation dataset. Additionally, we computed the Parameter Recovery Error (PRE) for each parameter for each of the twenty-five configurations, which is a measure of how concentrated the marginal is around the ground truth parameter value[49].

## Statistics and reproducibility
We compared MEA features of simulations generated with model parameters from the mode of the posterior distribution to MEA features of the corresponding experimental measurements. We performed statistical analysis using GraphPad Prism 5 (GraphPad Software, Inc., CA, USA). We first assessed the normality of the distributions using a Kolmogorov-Smirnov test. Given that normality of the features was often not ensured, we used non-parametric testing using a Mann-Whitney test to determine whether simulation and experimental features showed significant differences.

To investigate possible compensation mechanisms between model parameters, we generated 50 conditional distributions by holding all but two model parameters constant at values sampled from the posterior distribution and finding the distribution of the remaining pair of model parameters[17]. We then took 50 points from the resulting distribution and computed the Pearson correlation coefficient between the two model parameters, repeating this for every pair of model parameters. This resulted in 50 correlation coefficients per model parameter pair. To assess whether the average correlation coefficient per model parameter pair significantly differed from zero, we performed a one-sample $t$-test using GraphPad Prism 5. Normality was ensured using Shapiro-Wilk test.

To identify which parameter distributions were affected by drugs or disease, we compared the univariate marginals. First, we calculated the marginals using the MEA features of every network before and after drug application. We then took 50 samples from the marginals from each test and assessed whether the samples originated from different distributions using a Kolmogorov-Smirnov test using the Scipy stats package within a Python environment. This resulted in one $P$ value per network. The number of networks is indicated in the figure captions. Dynasore networks were from two independent neuronal and astrocytic batches and MK-801 networks were from three independent neuronal and astrocytic batches. For comparison between healthy and diseased networks, we averaged the MEA features per cell line per MEA (different independent neuronal and astrocytic batches) and calculated the marginals. The number of batches is indicated in the figure captions. Using 50 samples from each set of marginals, we performed a Kolmogorov-Smirnov test, resulting in a $p$ value per MEA.

## Reporting summary
Further information on research design is available in the Nature Portfolio Reporting Summary linked to this article.

## Data availability
All newly generated data is available on Gitlab with project ID 12956 (https://gitlab.utwente.nl/m7706783/SBI_MEA_model). The source data of the figures can be found in Supplementary Data 1.

## Code availability
The trained neural density estimator (NDE), the training dataset, the code to analyze experimental data and create the posterior distribution, the code to run simulations with the computational model, and the code to create the figures are available on Gitlab (https://gitlab.utwente.nl/m7706783/SBI_MEA_model). Simulations and analyses were performed in Python 3.8, primarily using the Brian2 package, version 2.4.2, and the sbi package, version 0.21.0.

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

## Acknowledgements

This work was supported by the Netherlands Organisation for Health Research and Development ZonMW; BRAINMODEL PSIDER program 10250022110003 (to M.F.). We thank Maurice van Putten for his invaluable support, expertise, and generous provision of the code to implement synaptic parallel computing for dynamic load balancing. Moreover, we thank Eline van Hugte, Marina Hommersom, and Nael Nadif Kasri for providing MEA recordings from patient-derived and genome-edited in vitro neuronal networks.

## Author contributions

N.D. conducted data analysis, designed and implemented the processing pipeline, and wrote the manuscript with input from all authors. M.F. and M.J.A.M.v.P. provided conceptualization and intellectual content. M.F. conceived and supervised the project.

## Competing interests

The authors declare no competing interests.
