## [Transparent Peer Review file · Communications Biology]

Automated inference of disease mechanisms in patient-hiPSC-derived neuronal networks

Corresponding Author: Ms Nina Doorn

Version 0:

Reviewer comments:

Reviewer #1

(Remarks to the Author)

This article proposes the use of simulation-based inference (SBI) – a machine learning framework - alongside a biophysical computational model previously developed and published by the authors to model the activity of neuronal networks cultured on MEAs and automatically identify disease mechanisms through detection of changes in model parameters.

The article aims to address an important issue in neuroscience: how to use and interpret MEA data to understand biological mechanisms underlying disease and patient-specific phenotypes.

The use of MEA technology has become very common in the field; however, most researchers only use standard recording metrics, such as mean firing rate, burst frequency and network burst number which most of the time are not reproducible across recordings and batches and are often inconclusive in the characterization of in vitro disease models. MEA data often contains intricate spatiotemporal patterns of neuronal activity that are difficult to interpret with traditional metrics and statistical methods.

Therefore, I appreciate the authors' work, and I agree on the scientific relevance of the proposed methodology.

The methods proposed by the authors are innovative. SBI is a powerful, machine-learning framework capable of handling high-dimensional, non-linear datasets. It has never been applied to MEA data. The authors show that SBI enables the inference of parameters from complex patterns of spiking/bursting activity and, by integrating MEA data with mechanistic biophysical models, it allows researchers to better understand the link between network activity and deeper biological processes such as ion channel dynamics and receptor function. This is something that is typically addressed only by using single-cell functional assays such as patch-clamp, RNAseq, or calcium imaging – very time consuming, invasive and laborious methods.

One of the major concerns is that SBI is a model-based approach, and its utility depends on the quality of the biophysical models used. The authors did a good job in validating SBI using both simulated and experimental data. They demonstrate the ability of SBI to identify ground-truth parameters and replicate experimental network activity. However, the authors acknowledge that the model may exclude relevant biological processes, such as inhibitory pathways or mitochondrial dysfunction. This limitation could significantly impact the accuracy and completeness of SBI predictions, especially for complex pathologies. It would be interesting to know if the authors plan to address these limitations and which future works should be implemented to improve the model and interpretation. For example, by adding parameters that better represent synaptic interactions, inhibitory effects, astrocytic influences.

The inability to validate the full posterior distribution limits confidence in the inferred parameters, and in the ability of the method to infer disease mechanisms. Posterior predictive checks are useful, but they do not replace direct experimental validation of inferred mechanisms, which is emphasized but not fully addressed. Include additional experimental evidence to validate the key predictions.

The lack of reproducibility across batches (e.g., potassium channel conductance in CACNA1A+/- neurons) undermines confidence in some conclusions. Due to the well-known variability of MEA experiments, I recommend performing additional replicates or explore/investigate better technical reasons for variability.

In some paragraphs, the text is sometimes difficult to follow, especially in discussing parameter space and statistical methods. Simplify descriptions where possible and describe SBI keeping in mind a broader audience.

Final recommendations: The article is promising but requires significant revision. If the authors can address the validation and reproducibility concerns and improve the clarity of the text, this paper is a good candidate for Nature Communications Biology. The novelty, methodological innovation, and potential impact align well with the journal's scope.

Reviewer #2

(Remarks to the Author)

1. Summary of Manuscript

In this paper the authors utilize a machine-learning approach named simulation-based inference (SBI) to estimate model parameters of a computational model of human induced pluripotent stem cells (hiPSCs) derived to neuronal networks on multi-electrode arrays. SBI can be used to identify the model parameters that can replicate the neuronal network activity. Here in this case, parameters that were finely tuned based off distributions to set the model parameters to replicate MEA neuronal network activity. Not only was this capable in a physiological normal condition, but SBI can also identify network activity that is sufficiently altered by pharmacologically induced changes, as shown with the case of a network with the addition of Dynasore. The network burst duration decrease was seen in the in vitro experiments and modeled in the simulations, and when multiple simulations resultant p-values were averaged the recovery time scale of the short-term synaptic depression was what was consistently concentrated at low values across the models. The paper also showed how if the pharmacological response was not consistent across networks, as the case with the MK-801 NMDA-receptor blocker, then only minor differences are detected. Hence this led the authors to declare that SBI cannot detect molecular changes that do not significantly affect the neuronal network pattern. SBI was also able to replicate disease mechanisms in genome-edited neuronal networks, in this case that for Dravet Syndrome (DS), Generalized Epilepsy with Febrile Seizures (GEFS+), and a case with a monoallelic frameshift variant in exon 8 of CACNA1A. For DS and GEFS+, the SBI was able to, over two batches, detect shifts to lower values in the noise, sodium conductance, and potassium conductance. As weak as an increase in short-term depression which was noted to recently in the literature be associated with GEFS+ and DS networks. For the altered CACNA1A, over three batches, the SBI found significant changes toward lower values in the noise, AMPA conductance and connectivity: in one batch a significant shift to lower values in potassium conductance. Overall, this paper sets the framework on how SBI may be used as a method to detect disease mechanisms and alterations in MEA neuronal networks.

2. Overall Impression of the work

This work is novel in applying the SBI method to MEA networks to get extract the dynamics of the network specifically though the changes in the model parameters. It is an interesting approach to see how altering model parameters can help unravel the changes that are occurring within neuronal network. This paper shows that the process to do is still a bit in its infancy as there are still limitations such as the need for posterior distribution using multiple MEA measurements, reliance on model completeness and computational expense. However regardless of these limitations SBI allows for a robust method to analyze a model's parameters in relation to altered conditions of the neuronal network.

3. Specific comments, with recommendations for addressing each comment

1. Minor comment, a quick fix for Figure 4 caption C is missing a "ns" before Average $P > 0.05$.

2. In lines "93 We analyzed the resulting simulations by computing 13 MEA features that capture the 94 most important characteristics of the network activity and are often used in MEA literature (see 95 Methods and Table 2 for details)." Here perhaps or in the methods include the literature or justification on why these were the 13 parameters selected and others were excluded (such as how was done with the high correlations between the correlation coefficient between binarized spike trains but include other examples and their respective justifications rather than generalized that too many parameters cause too many variation).

3. One comment In lines "373 it is crucial to compare neuronal networks cultured on the same MEA and measured simultaneously, as differences in astrocyte and MEA batches can introduce confounding factors." The computational model and study include no mention of astrocyte modeling and therefore it may be important to make note of this and how it may still be unknown if SBI would be able to test for mechanisms including astrocytes as this analysis was done primarily on a neuronal network model. It may be important to note this that this analysis of certain parameters and their dynamics do depend on the model and its parameters that are identified.

Reviewer #3

(Remarks to the Author)

Disclaimer: I am not familiar with hiPSC-NNs data, but have practical experience in the analysis of in-vivo and in-vitro mammalian electrophysiological signals. Where I may have misunderstood the authors' intent, I encourage them to point it out in their responses, and if applicable, to kindly understand an implicit need for a clearer explanation to makes the work accessible to a wide audience of neuroscientists from different subfields.

Overall impression

The authors estimate the parameter values that allow a biophysical model of excitatory neuronal networks to closely reproduce multielectrode recordings of the activity of a network of cultured **hiPSCs. They claim that their approach using simulation-based inference yields estimates that plausibly track pharmacological and gene-editing interventions and even allows to identify key disease mechanisms.**

The present study extends to hiPSC-NNs the recent trend of using SBI for reliable inference of inverse problems in

neuroscience, which in my view is both timely and relevant, given the claimed diagnostic potential of hiPSC-NNs. The text is well articulated (in particular the introduction does an admirable job of presenting the relevance of the problem, typical approaches, shortcomings and the advantages of SBI with scant use of machine-learning jargon), the schematics and data displays are well organized and clearly presented, and the results are critically discussed with transparency about the limitations of the approach.

With some technical improvements to buttress the authors' use of sbi, with which I have chiefly concerned myself for this review, I believe this will become an influential study, since it exemplifies how to interrogate hiPSCs-NNs with state of the art inference techniques and presents findings in a balanced and accessible way to neuroscientists at large. Even if the SBI approach "just" accelerates hiPSCs-NNs fingerprinting by providing reliable parameter estimates otherwise only accessible via direct experiments, this constitutes a very significant speed up for research both on neuronal network as complex systems and of their spontaneous activity as a subtle biomarker for clinical conditions.

Major comments

1. ****Research design and scope of conclusions****. While pharmacological and gene-editing interventions can be performed directly on object of the simulation, namely the cultured tissue, a disease affects the organism from which the hiPSCs are sourced. Cultures of a diseased individual's hiPSCs-NNs act as a proxy for a much more complex neural system; it is crucial to exercise caution (and warn the reader) about drawing conclusions for living human brains from parameter values obtained with SBI (indirection #1) based on comparing simulated electrical activity (indirection #2) with experimental multielectrode recordings (indirection #3) of in-vitro tissue (indirection #4). As long as the cultured tissue and the simulator are only used to fingerprint activity based upon which SBI can flag potential conditions for further investigation in vivo, that is safe territory. This limitation is shared with similar research, so it is only a matter of finely acknowledging where appropriate, for example:

1. clarifying in L316-319 that the simulator model is designed for a closed (in complexity and spatial extent) cultured system, and that statements pertaining to disease, i.e. about the open-system, in-vivo network, are additionally uncertain, reminding that that both the simulator and the direct measurements with which parameter inferences are compared refer to cultured tissue
 2. while L19 in the abstract is clear in this respect, I am uncertain about readers expecting too much in terms of elucidation of in-vivo mechanisms based off L10-L13 and L20, L22. I recommend to explicitly separate the statements on disease mechanisms from the others and use more cautious language
 3. L34 furnishes, in my view, an example of adequate language, with L35-36 situating the work as part of in-vitro based hypothesis making about in-vivo systems, which the paper intends to substantially accelerate
 4. see Minor comments > Discussion > Limitations > 1 for more specific suggestions
2. ****Limitations common simulator-based research****. The following limitations are common to research based on simulators, and as such they do not challenge this study specifically but rather as part of an entire approach to researching complex systems. In fact, this study should be commended for adopting SBI and thus entirely avoiding a third common limitation: the inability of most popular scalable inference techniques to identify multiple parameter sets compatible with the observations and provide meaningful error estimates.

1. The simulator may not adequately represent the system ("model misspecification"). The target here should be to ensure the simulator is well specified wrt. the in-vitro system; the complexity of the in-vivo tissue would likely be out of the scope of present SBI techniques in terms of number of parameters involved. Since the literature on diagnostics for model misspecification in the SBI field is still nascent [1, 2, 3], and the authors use a previously published and validated forward model for hiPSCs-NNs on a best-effort basis, I recommend, that authors, at their discretion and for increased trust,

1. run checks proposed in [1],
 2. or, if using embedding networks (see below and in minor comments to Methods), that they use those proposed in [2]. This might represent a considerable effort (albeit less than for those suggested in [1]), but, in any case,
 3. authors should refer to model misspecification by name and outline what steps appropriate to their system they or others could undertake in future work to diagnose potential misspecification
2. The parameter inference in SBI might be based on a reduced set features of the measurements due to investigative focus, interpretability concerns or computational limitations, so that inference may be underconstrained or not attend to key dynamical aspects of the system (insufficiency of "summary statistics"). In this study, the authors reduce one-minute time series to 13 time and frequency-domain scalars (e.g. rates and inter-event intervals of spikes and bursts), but it is unclear if these can fully capture the underlying dynamics (cf. cautionary example in [4]), even if the resimulated traces reproduce the bursting behaviour relatively well. To strengthen their study, I recommend to compare current results from hand-crafted features with those derived from employing either

1. an unsupervised technique for dimensionality reduction, such as PCA (as demonstrated in [4]), or, better yet,
 2. a data-driven approach to summary statistics, by using an embedding network (something that is enabled by the author's choice of using neural posterior estimation, NPE). Relevant examples of this approach are [5, 6]; both using convolutional networks and some form of transfer learning and the second specific to neuroscience time series. See minor comments to the Methods for more concrete recommendations.
 3. Regardless of the path taken, authors can compare results as outlined in [4, Fig 4] with PRE (parameter recovery errors) besides PPC; this reference also proposes OVL as a means to compare posterior distributions; see also [11] for feature importances based on KL distances of posteriors.
3. ****Limitations in this study's use of the SBI approach, and its reporting for review and reproducibility efforts.****

1. Authors claim to use a mixture-density network (MDN) as posterior density estimator, but the published code in the accompanying repository at https://gitlab.utwente.nl/m7706783/SBI_MEA_modelis is quite incomplete, so that the interested reader cannot find how the MDN is parameterized, or, in fact, any of the training code at all. This is a severe shortcoming and for me as a reviewer who desires to understand the work, a red flag for publication (a false appearance of transparency is worse than the lack of transparency). I recommend that in a further submission:

1. The accompanying code should be complete. Publishing just the network weights, despite PR attempts by e.g. Meta with their Llama models, falls short of fully open sourcing the model, which is necessary for proper reproducibility and hence for scientific discourse. Weights are helpful, but the full training protocol is essential and, together with the measured data and simulation results (again, better the simulator source code), allows any interested party to recover the weights just with some computational effort.

2. The density estimator should also be briefly described in the Methods section of the main text, indicating number of layers, GMM components, etc,

3. The peculiar choice of an MDN should either be quantitatively justified, or a more powerful density estimator should be adopted (see e.g. the seminal publication [16], cited by the authors). For context, I surveyed related applied literature wrt. choices of density estimators and found 10 references ([4, 5, 6, 7, 8, 10, 12, 13, 15, 16]) preferring normalizing flows, most of the time a Masked Autoregressive Flow (MAF; [the default for NPE in the sbi toolkit](https://github.com/sbi-dev/sbi/blob/f2c1cc3d623f8a8a71eb3c132357b2d376314b73/sbi/inference/trainers/npe/npe_base.py#L51)) and occasionally a Neural Spline Flow (NSF; performing better than MAF in the SBI benchmarking paper [18, Appendix H.5]), vs. a single reference ([9]) adopting a MDN.

2. When conducting inference, authors decide to leave some parameters fixed, adducing that the interpretation would be further complicated by the washed-out marginals they expect in case of higher parameter space dimensionality. I believe these improvements could be considered:

1. Provide model-specific, neuroscientific rationale for why the authors decided to consider some specific parameters as fixed

2. (effortful) If the computational performance of the potential new density estimator allows, leave more (ideally all) now-fixed parameters free for adjustment within sensible prior ranges; After inference, those parameters for which the range of variation is (fractionally) small can be fixed for discussion and interpretation. That is, authors could already in Fig. 1B use conditional posteriors, where those parameters that have “uninteresting” ranges act as conditioners. This makes sense insofar that it allows SBI to find potentially better combinations of parameters in the enlarged space, yet authors can focus the discussion on (potentially) the very same parameters they focus on today (or others, depending on the outcome).

[1] [[2412.15100] Tests for model misspecification in simulation-based inference: from local distortions to global model checks](<https://arxiv.org/abs/2412.15100>)

[2] [[2406.03154] Detecting Model Misspecification in Amortized Bayesian Inference with Neural Networks: An Extended Investigation](<https://arxiv.org/abs/2406.03154>)

[3] [[2209.01845] Investigating the Impact of Model Misspecification in Neural Simulation-based Inference](<https://arxiv.org/abs/2209.01845>)

[4] [Methods and considerations for estimating parameters in biophysically detailed neural models with simulation based inference | bioRxiv](<https://www.biorxiv.org/content/10.1101/2023.04.17.537118v1.full>)

[5] [[2412.02437] Reproduction of AdEx dynamics on neuromorphic hardware through data embedding and simulation-based inference](<https://arxiv.org/abs/2412.02437>)

[6] [Real-Time Gravitational Wave Science with Neural Posterior Estimation | Phys. Rev. Lett.](<https://journals.aps.org/prl/abstract/10.1103/PhysRevLett.127.241103>)

[7] [Uncertainty mapping and probabilistic tractography using Simulation-Based Inference in diffusion MRI: A comparison with classical Bayes | bioRxiv](<https://www.biorxiv.org/content/10.1101/2024.11.19.624267v1.full>)

[8] [Simulation-based inference on virtual brain models of disorders | IOPscience](<https://iopscience.iop.org/article/10.1088/2632-2153/ad6230/meta>)

[9] [Pathological cell assembly dynamics in a striatal MSN network model | Frontiers](<https://www.frontiersin.org/journals/computational-neuroscience/articles/10.3389/fncom.2024.1410335/full>)

[10] [Amortized Bayesian inference on generative dynamical network models of epilepsy using deep neural density estimators | ScienceDirect](<https://www.sciencedirect.com/science/article/pii/S0893608023001752>)

[11] [Combined statistical-mechanistic modeling links ion channel genes to physiology of cortical neuron types | bioRxiv](<https://www.biorxiv.org/content/10.1101/2023.03.02.530774v1.full#sec-9>)

[12] [Bayesian Inference of a Spectral Graph Model for Brain Oscillations | bioRxiv]

(<https://www.biorxiv.org/content/10.1101/2023.03.01.530704v2.full#F5>)

[13] [The virtual aging brain: a model-driven explanation for cognitive decline in older subjects | bioRxiv]
(<https://www.biorxiv.org/content/10.1101/2022.02.17.480902v2.full>)

[14] [Fast inference of spinal neuromodulation for motor control using amortized neural networks | IOPscience]
(<https://iopscience.iop.org/article/10.1088/1741-2552/ac9646/meta>)

[15] [Inferring Morphology of a Neuron from In Vivo LFP Data | IEEE Conference Publication | IEEE Xplore]
(<https://ieeexplore.ieee.org/abstract/document/9441161>)

[16] [Training deep neural density estimators to identify mechanistic models of neural dynamics | eLife]
(<https://elifesciences.org/articles/56261>)

[17] [Fast ϵ -free Inference of Simulation Models with Bayesian Conditional Density Estimation | Neurips]
(<https://proceedings.neurips.cc/paper/2016/hash/6aca97005c68f1206823815f66102863-Abstract.html>)

[18] [Benchmarking Simulation-Based Inference | AISTATS](<https://proceedings.mlr.press/v130/lueckmann21a.html>),
([Appendices](<https://proceedings.mlr.press/v130/lueckmann21a/lueckmann21a-suppl.pdf>))

Minor comments

Introduction

1. L45 and the network CONNECTIVITY (looking e.g. at Table 1, it seems that the network is represented here just by the synapses and the wiring probability and spatial patterning)
2. L51 with -> using, as, via
3. L59 one could add in parenthesis, to better convey how damning and inescapable this phenomenon is, the term of art in machine learning: “curse of dimensionality”
4. L60 there is a chance to introduce the concept of amortization, e.g. “(i.e. they are non-amortized)”
5. Ref. 18 is not immediately applicable to the SBI techniques in this paper and can confuse the reader looking for more detail. Ref. 19 is more relevant (using normalizing flows), but should be replaced by a different one (e.g. [17], using MDNs) if the authors present good reasons to keep using the MDN.
6. L69 not using simulations “only”; since SBI is a Bayesian framework, priors provide a structured way to leverage existing knowledge about the parameters, and help speed up inference; the authors likely intend “only” to convey that only sampling, and no evaluation of a likelihood function is needed. Unfortunately, there is no preceding discussion of likelihood-based methods, so this implicit opposition may be confusing for readers as it was for me.
7. L80 does not mention the gene-editing data
8. L75-85 mix tenses

Results

SBI correctly identifies...

1. L91-92

1. “i.e.” seems to imply that a (any) prior distribution *is* a uniform distribution. I advise to give due treatment to the idea of a posterior in the introduction (see above) or in a separate sentence, where authors can justify the use of their chosen prior (in absence of specific reasons, as a customary approach often taken when approaching Bayesian estimation with limited information, i.e. “this parameter must be positive” already fixes one side of the interval, and some physical / physiological reasoning easily fixes the highest allowable value of the parameter”).
 2. the term “box prior”, used in the sbi library, conveys perhaps better the situation when the uniform distribution over a range is, in fact, multidimensional, i.e. uniform over the cartesian product of several 1D intervals.
 3. Fig 1 panel A1 confusingly depicts a prior that is *not* a box prior as discussed above. Since panel A2 introduces details of the actual simulator (instead of remaining at the schematic level as a “black box”), A1 should be at least not contradictory wrt. the text and depict a square of equal probability. Adding “(box prior)” or “(uniform)” to the panel title could support the connection between text and diagram. Similarly for the caption: since in 2 simulations are performed using *our biophysical model*, in 1 the parameters should be drawn from e.g. “within plausible ranges (see Table 1)”
2. L92

1. double parenthesis; use citep or reorder the sentence to facilitate reading (ideally, do not present prior in a parenthesis, since it is an important notion)
2. no mention is made at this point of why some parameters are considered subject of investigation while others are fixed with, effectively, infinite precision; authors can defer explicitly to the discussion.
3. L94 - recommend introducing the term of art: summary features/summary statistics: “13 MEA features that capture the most important characteristics of the network activity (summary statistics)”. The superlative “most important characteristics” is not warranted in a parameter inference context, and I would contest that even from a signal analysis point of view these particular features are the most important. Which are most important can only be decided in a data and task-dependent manner (see discussion of the Methods section below). This remark doesn't negate the commendable effort of the authors in

- using many features that cover both time and spectral domain aspects of the recording and are community-vetted
4. L99-100 - compatible THE PRIORS and experimental observations. Further, a distribution should not be called a space, especially one with infinite tails, as, effectively, every parameter combination would be "compatible with", albeit most of them not "probable under", the observations. The posterior distribution has been already introduced in L71; the concept can be directly used
 5. L102 - simulations mimicking data? (I take data here to be measurements)
 6. Figure 1A6, caption: the \rightarrow one, per \rightarrow for each. For clarity, authors should stress early on that a recording is conceptualized as a single observation, and/or be careful when using "observations" in plural, since it may cause confusion: yes, they are observationS in time, but SBI sees them as a single observation, a 13-component vector.
 7. Figure 1A6, caption: mixes tenses wrt. 1A1-1A5 and 1A7
 8. Figure 1, caption C: separate words in rasterplot (as line plot or bar plot, though admittedly, scatter plot is used also in its single word variant)
 9. Figure 1, caption D: please set $n=10$ in math mode for proper spacing, here and elsewhere in the manuscript
 10. L107 - use m-dashes for the parenthetical expression (<https://www.merriam-webster.com/grammar/em-dash-en-dash-how-to-use>), and do not write a dash between ground and truth, as this would join those words into an adjective
 11. Fig 2C - is one minute enough to estimate low-uncertainty summary features of MEA activity, in particular those involving ISIs of low-firing cells? This really depends on the observed rates. Note one advantage of summary features is that they allow, in principle, to use different-length recordings (irregular data)
 12. Fig 2A, caption: $\$p\$$ -value or P value? I prefer downcase $\$p\$$ -value, as we're talking about a realization, not a random variable (unless Nat Comm Bio stipulates otherwise in their instructions to authors). See discussion at <https://stats.stackexchange.com/questions/871/correct-spelling-capitalization-italicization-hyphenation-of-p-value>
 13. Fig 2A, and all subsequent figures displaying single values as dots---please use a smaller dot size and jitter the dots horizontally (see e.g. seaborn's stripplot or swarmplot functions). Otherwise, dots occlude each other, e.g. in the bar plot #Fragments/NB there are eight dots over each bar, which means that simulations do not show almost any variability in this metric
 14. Fig 2A, bar plots, left bar dots - even within a single subject, the variability of summary features seems very substantial . This makes me wonder about using a single control subject for comparison with diseased individuals. Comparisons are made always on inferred parameters - but how distant are the 13 summary features themselves? I would welcome insights from the authors on this issue

SBI can identify the entire landscape ...

1. L114 "parameters" cannot be a subject to the verb simulate---they can enable somebody to simulate
2. L119 to check/EVALUATE reproducibility (no THE)
3. L122 the batchwise variability for a single individual makes me wonder whether the variability across individuals is not so large that the notion of a "control" subject loses its meaning
4. L136 and L148 I advise to use not weak resp. high correlations but weak resp. strong associations, since the eye will perceive associations beyond just linear (see comments on Methods about quantification)
5. L141 the remaining ONE or TWO model parameters could take to still REproduce the desired network activity. This resulted in conditional posterior distributionS (since there are distributions conditioned both to $k-1$ and $k-2$ parameters)
6. L145 vary within a limited range ___and strongly constrain each other___ in order for simulations to match observations
7. L149 with/OVER many possible CONDITIONING configurations/SETS
8. L151-152 I find the structure of the sentence confusing. Presumably what is meant by the number of synapses is the parameter Conn%. It would be good to reuse the terminology given in Table 1 (probability of connection), even if the number of synapses is related (I am also unsure if the model contemplates multisynaptic connections). Furthermore, the two comparisons should be separate: even if both the AMPA and NMDA conductances were strongly negatively correlated to the probability of connection
9. L154 consequently, an experiment with NMDAR blockers and measuring short-term synaptic depression on this system might be proposed in the discussion for electrophysiologists to test the predictive capacity of the approach

SBI can identify pharmacological targets ...

1. L171 synaptic vesicle recycling (COMMA) thereby
2. L176-L177 consider metrics such as e.g. OVL (see above) for quantification at the level of the posterior
3. L178 simulations with the mode of EACH [not BOTH] posterior distribution (...) appear highly similar to the CORRESPONDING
4. L184 I am not a statistician, but it is unlikely that averages of a nonlinear measure, such as a $*p*$ -value are very meaningful, and they are most certainly not a p-value that can be compared (dashed line) to typical thresholds of significance such as 0.05. I can think of two solutions:

1. explore multiple K-S tests (see <https://stats.stackexchange.com/questions/35461/is-there-a-multiple-sample-version-or-alternative-to-the-kolmogorov-smirnov-test> for references)
2. or, instead of spending effort in sophisticated averaging of $*p*$ -values (e.g. as outlined in <https://www.semanticscholar.org/paper/Combining-P-Values-Via-Averaging-Vovk-Wang/29c9b82c9fd88d2e9dba9ad79301a281916aee6>), I suggest if the number of $*p*$ -values is reasonable, to use a simple per-parameter scatter plot for Fig 3D and similar (i.e. use the jittering approach described above). The distributional properties of the $*p*$ -values are more faithfully represented than with typical bar and error bar plots (in particular, extreme values can be spotted), and they can all be individually compared with significance thresholds just fine. The interpretation in the text does not need averages either.

3. In any case, for readability, you may want to plot the range [0.0 - 0.05] in an inset with enlarged scale, or use a logarithmic scale for the y-axis, in which case you can add dashed lines at *, **, ***, etc significance levels to guide the eye, relevant to Fig 4.

5. L198 the conclusion is not complete: perhaps the summary features do not change significantly or consistently but the underlying neuronal network activity pattern does. Besides acknowledging this possibility (which authors do perfectly in L262 in the discussion), there are two ways to improve here: one is to quantify the changes induced by MK-801 at the level of features; the other is to (as suggested above) use more comprehensive activity fingerprinting, either via PCA and retaining many components or even better, as suggested above and explained below, by adopting an embedding network

SBI pinpoints known disease mechanisms ...

1. L205 authors say "because we previously observed that different batches affect SBI results..." it is not clear how much observations from different batches already differ at the level of the 13 input features.

1. Fig S2B and Fig 3: the variability across batches is transparently presented in order to contextualize the putative disease-driven or genetic-manipulation-induced variability, but what about the inter-individual variability? Could a second control subject could help understand this driver of variability? The drug experiments are both based on cell lines from the same subject, so this comment does not apply there

2. Fig. 4C and Fig. 4F --- same comments as above apply to aggregation of *p*-values, but here it is really critical to reconsider the approach because the stars may give the impression that the average *p*-value is a *p*-value, which is not

3. L231 same comment as above where it is not clear that number of synapses stands for Conn%

Discussion

This section compares the changes observed in SBI-estimated parameters after a number of interventions both to known mechanisms and direct, independent experimental findings in vitro, and makes an overall convincing case for the authors' use of SBI and for the field to adopt it

1. L239 the phrase can be improved to not use a generic word like utilizing or using ("MEAs allow quick collection of ephys data from hiPSC-NNs"). If you insist on utilizing, however, I find these stylistic remarks appropriate: <https://www.merriam-webster.com/grammar/is-utilize-a-word-worth-using>

2. L245 for the expressions: RNA sequencing, time consuming: don't join words with a dash; they would receive one if they preceded a noun, such as experiments

3. L253 WHEN using Dynasore (...) [otherwise it seems that SBI was using Dynasore]

4. L262

1. "clearly" is obvious (no analysis procedure can speculate phenomena that leaves no trace in their input data) so that it might invite the impression that there is a limitation of SBI at play. But this is not the case - SBI is flexible to accommodate more comprehensive and more data-driven summaries of observations as inputs -- as described above. I suggest to rewrite along the lines "parameter inference (and hence SBI) can only identify changes that are already clearly and consistently reflected in the descriptors of system behavior that constitute its input"

2. "consistently" begs the question of the relative impact of batch-specific factors (nuisance factors) wrt. all others on the network activity. It would aid understanding to quantify how interventions impact MEA features vs. just drawing from another batch under no intervention (same for drawing from another control individual, same batch)

5. L273 - since forecast has connotations relating to time I suggest to use estimate instead (there is no before/after the disease with a comparison in the same individual); further, there is no "increase" because there is no temporal ordering ---it would be more appropriate to write that SBI estimates a higher STD under this or that intervention ... Accordingly, predictions -> inferences in L275

Limitations and advantages

1. L317 - here misspecification is discussed in a potentially confusing way. As highlighted in Major comment 1, it must be taken for granted that the model is very strongly misspecified with respect to the in-vivo network. Yet, this does not detract from its main conclusions 1) that SBI is a great advancement in parameter inference in biology, and 2) that results leveraging the authors' simulator are largely in agreement with known mechanisms and direct, in-vitro measurements on comparable model systems. In line with the initial comment, I suggest to clearly state that one should be cautious in drawing conclusions from an entirely in-vitro framework for the living system, insofar as the hiPSCs-NNs are a relatively simplified model for it, and that in any computational investigation of hiPSCs-NNs the computational model (the simulator, here) may not fully represent the dynamics even if they are already much simplified in comparison to in vivo. So the misspecification that is not trivial and matters to highlight is that of the simulator and the in-vitro model, not the obvious and likely insurmountable one if the simulator was attempting to model the in-vivo behaviour, which is not.

Yet, the results speak for themselves and support the publication of this paper, with all the provided caveats and potentially with more work to quantify misspecification, leading to an improved model in the future and warrant a nuanced cautious recommendation to widely adopt SBI in the community.

2. L324 I am unconvinced that a more involved interpretation is a price to pay for potentially fitting a misspecified, because overconstrained (via fixed parameters), model, and I offered a suggestion elsewhere. Regarding the computational burden of the simulator, I do agree it represents a practical problem as parameter numbers increase, but for the sake of future users of their simulator, I would like the authors to give representative orders of magnitude of simulation times in the Methods.

3. L328 this might be a plausible argument in terms of pathophysiological detection or classification, but then we don't need a mechanistic simulator and sophisticated inference engine: just the embedding network discussed elsewhere coupled to a classifier would do. The danger of a misspecified model is that some parameter not causally linked to the particularities of an observation will be tuned to best match it, because the really responsible parameter is not free to vary.
4. L337 I appreciate the introduction of additional features in search of a sufficiently sensitive set
5. L341 sounds like a concrete, actionable, feature request for the simulator or the MEA summaries. If the current simulator cannot match the in-vitro typical signal-to-noise ratios (SNR) in silico, it should be structurally improved; alternatively, if it is sufficiently expressive already, then, the SNR might constitute a suitable 14th handcrafted MEA summary feature?
6. L344-348 note that the sbi library bundles more and more recent posterior checks, such as TARP (complete list here: <https://sbi-dev.github.io/sbi/latest/#diagnostics>)
7. L353-368 this paragraph articulates very clearly why SBI is the most compelling approach for parameter inference on computational models of biological function, and why this paper deserves publication at this venue. Consequently, I find the last sentence (L367-368) makes a too conservative summary of the paragraph and the author's comprehensive work. Based on this and the rapidly growing body of applied literature across the most diverse fields of science, SBI (not just posterior analysis, the posterior may be inferred in other ways) not just "could" be a "plausible addition" to the MEA analysis pipeline, but it constitutes the most appealing of the (already more compelling) Bayesian approaches, sometimes the only plausible in high-dimensionality parameter spaces.

Recommendations

1. L374 note that if those confounding factors could be mechanistically modeled within the simulator, it would be possible either to co-infer e.g. the astrocyte count or where experimentally feasible, to measure them for each batch and have batch-specific simulators to account for batch-local conditions.
2. L379 a SUFFICIENT NUMBER/QUANTITY of batches
3. L381 I believe (see above) that averaging p-values is unsound statistical advice and that it suffices, in absence of a more principled procedure, the authors should limit their recommendation to drawing conclusions based on the analysis of a sufficiently representative amount of batches
4. L383 SBI-identified (see above)
5. L384 this is meanwhile not anymore correct, see above

Methods

Computational model

1. The numerical integration of the model is not discussed. If using forward / exponential Euler like in previous publications (e.g. Doorn et al. 2023, Stem Cell Rep.), how has it been ensured that there is no accumulation of error over the integration time? Is there an option in Brian2 to employ a higher-order method, e.g. from the (stochastic) Runge-Kutta family with better stability properties?
2. The citation to the synaptic-parallel computing engine is obscure: it is unclear whether the use of this (custom?) platform is merely for speeding simulations up, or if reproducibility of the work with standard computers would be hampered. It is difficult to understand the platform by reading the appendix of an astrophysics paper (reference 40). Could the authors briefly indicate what prompts the decision to use the platform, and if any specifics of it are liable to introduce differences with a standard serial architecture? The published code, upon cursory inspection, does not show traces of the use of any non-standard hardware

Preprocessing

1. While the Butterworth filter, being an IIR filter, is fast and can be used online, FIR-type filters such as Parks-McLellan are preferable in situations where the whole signal is available at once, such as is the case here (but FIR filters have border effects, so that the beginning and end of the recording are not directly usable). Another reason to promote FIR filters is how easily they compose with smoothing kernels, see next point.
 2. For more elegant signal detection, consider merging the low-pass filter used initially with subsequent phases of time-domain aggregation/averaging
1. computation of firing rates over bins is equivalent to convolution with a square-window convolution kernel (which is spectral properties)
 2. smoothing of firing rates with a Gaussian for fragment detection is yet again another low-pass filter with a Gaussian kernel
 3. ... so you can just design composite filter kernels and use the once for the respective features
 4. ideally the peak detection algorithm parameters should be discovered by systematic comparison with human-labeled spike trains (e.g. by training a segmenter or classifier model). But at that point it might be better to use the end-to-end approach discussed below
 5. the composite-filter approach can also be done on 2D to account for spatial features of the MEA firing pattern
 6. But, overall, the main weakness is the selection of summary features and the elaborate hand-calibrated detection, with obscure settings such as 1/16th peak height and 1/20th prominence, etc, which emerge from tedious visual inspection by an expert. What I would like to see to strengthen results, is an end-to-end approach, where the entire signal is processed by a convolutional network (1D) over the spike-time series, potentially with skip connections, but has electrode exchange invariance (i.e. it handles electrodes as a set). As reasoned above, a number of summary features based on more or less instantaneous rates emerge naturally from the hierarchical-convolutional approach. The approach can be applied for the ISI time-series just as well (or to a frequency-domain representation of short-term rates), and the outputs of these summarizing

networks can be concatenated for further downstream processing

1. if the spatial structure of firing times and intervals (e.g. for SD ISI elec) might be important, this should be done over 2D and there is no need to introduce exchange invariance among electrodes; if the only scale relevant to the phenomenon is of the size of the MEA array ("network" events), this will be picked up by the convolutional network by using appropriately wide convolution kernels
2. researchers in [6] used an autoencoder to find summary statistics to then feed to SNPE. They pretrained it by forcing it to reproduce synthetic data and then refined the encoder part together with the training of the NDE (a MAF). Sections II-C and II-D contain the details (note I interpret they want to say 'jointly' instead of 'in parallel'). This could be a useful template to follow in order to get data-driven summary features. The library sbi already has all the facilities in place to use an embedding network
3. this represents some additional effort, and not doing it, in my view, is not a showstopper for publication, but a more comprehensive and critical engagement with the selection and extraction of summary features would provide a qualitative leap in the quality of the paper and trustability of results, so I would also value any efforts to render the current selection more convincing in the absence of a more structured approach -- authors discuss critically the limitations of hand-crafting summary features indeed, but they could also explore the sensitivity of results against the inclusion of redundant features or the removal of (presumed) key ones
3. L484 - the authors initially trained the NDE with additional features that they then argue can be discarded because based on a high value (0.96) of correlation (a linear measure of association), they see those as redundant. However, I believe that the stronger argument, given that they already have trained the NDE, would be based on the distributions (or at least their modes) of the SBI-derived parameters. Adding redundant inputs should not affect those significantly. To what extent is this the case?

Conditional correlations

1. To detect the presence of complex relationships in conditional marginal distributions, authors would need to go beyond linear associations as captured by correlation coefficients and compute measures of mutual information (see third row of first figure in <https://en.wikipedia.org/wiki/Correlation>). This can only strengthen their argument in general, even if in most of the conditional marginals shown the association is relatively well-captured by a linear relationship, i.e. based on data shown, I do trust the overall thrust of the argument

Good work! I enjoyed reading the paper.

Kind regards, Álvaro Tejero-Cantero.

Version 1:

Reviewer comments:

Reviewer #1

(Remarks to the Author)

I have carefully reviewed the revised version of the manuscript along with the authors' point-by-point responses. I am pleased to see that all reviewer comments have been thoroughly addressed.

The authors have made substantial improvements to the clarity and overall presentation of the paper. In particular, I appreciate the enhanced discussion of the data, which has added important context and improved the interpretability of the results. The addition of supplementary data further strengthens the manuscript and supports the main findings effectively.

Moreover, the authors have clearly articulated the challenges associated with their work and have thoughtfully proposed directions for future research. These additions enrich the discussion and provide valuable perspective to the reader.

In summary, I find the revised manuscript to be significantly improved and well-prepared for publication, and I recommend acceptance.

Reviewer #2

(Remarks to the Author)

I have read the rebuttal letter, and I agree with the changes they had added based on my comments.

Reviewer #3

(Remarks to the Author)

First I want to thank you, the authors, for an excellent rebuttal text, that so much facilitated my review.

The revised manuscript is, to put it shortly, impressive. The authors introduce multiple new analyses in support of their original contribution, some of them representing a very substantial effort and all of them relevant to substantiating the discussion we have been having in a satisfactory manner.

A particularly salient example is the training of a dedicated embedding network. This analysis offered a valuable negative result, which is worthy of sharing with the SBI community and discussing in the appropriate venues.

While I believe the authors correctly appreciate that including Rebuttal Fig. 4 in the article might sidetrack the neuroscience-minded reader of the present journal, I would suggest to them, and the Editor, to consider adding that Figure and relevant paragraphs in Appendix or, at their discretion, briefly mention in the text how their experiments with embedding networks (necessarily limited!) resulted in inconclusive, washed-out posteriors at an elevated inference cost.

The interest of reporting this effort (whether in the present publication, or elsewhere) lies in how it might spark a discussion in the SBI community about practical approaches to combine the strength of expert-vetted features and the flexibility of neural network-based feature identification, which is a theme I have thought about (SBI takes the path not of replacing mechanistic modeling with expressive, but blind, neural networks, but of using the latter in the service of our hard-won understanding of systems as condensed into mathematical laws and calculated using simulators. Similarly, we should strive for neural networks not to replace summary statistics, something that, as Fig. 4 intimates to us, can be a hard and costly program, but to improve them.

The MMD assessment, newly computed PRE values, “ablation” analysis of the impact on posteriors of burst-related features, energy distance analysis, and the clarifications of intent and language, all further contribute to the solid and rigorous piece of work.

In sum, the paper in current form represents a strong contribution to the field of MEA analysis and an exemplary practical application of SBI techniques to the advancement of science, with a clear and pedagogical message about interpretational pitfalls and limitations. I sincerely hope that it will be widely read by both communities and that future applied SBI work takes inspiration from it.

Nina Doorn
Drienerlolaan 5
7522 NB Enschede
The Netherlands

Telephone: +31611398957
n.doorn-1@utwente.nl

Enschede, 03-04-2024

Please find the point-by-point rebuttal to the reviewers' comments below. All manuscript changes are highlighted in red.

Reviewer #1

The article aims to address an important issue in neuroscience: how to use and interpret MEA data to understand biological mechanisms underlying disease and patient-specific phenotypes. The use of MEA technology has become very common in the field; however, most researchers only use standard recording metrics, such as mean firing rate, burst frequency and network burst number which most of the time are not reproducible across recordings and batches and are often inconclusive in the characterization of in vitro disease models. MEA data often contains intricate spatiotemporal patterns of neuronal activity that are difficult to interpret with traditional metrics and statistical methods.

Therefore, I appreciate the authors' work, and I agree on the scientific relevance of the proposed methodology.

The methods proposed by the authors are innovative. SBI is a powerful, machine-learning framework capable of handling high-dimensional, non-linear datasets. It has never been applied to MEA data. The authors show that SBI enables the inference of parameters from complex patterns of spiking/bursting activity and, by integrating MEA data with mechanistic biophysical models, it allows researchers to better understand the link between network activity and deeper biological processes such as ion channel dynamics and receptor function. This is something that is typically addressed only by using single-cell functional assays such as patch-clamp, RNAseq, or calcium imaging – very time consuming, invasive and laborious methods.

We appreciate the time and effort the reviewer has taken to evaluate our manuscript. We are grateful for the thoughtful and constructive feedback, which has helped us refine and strengthen our work. Below, we address each of the reviewer's comments in detail and outline the corresponding revisions made to the manuscript.

1.1. One of the major concerns is that SBI is a model-based approach, and its utility depends on the quality of the biophysical models used. The authors did a good job in validating SBI using both simulated and experimental data. They demonstrate the ability of SBI to identify ground-truth parameters and replicate experimental network activity. However, the authors acknowledge that the model may exclude relevant biological processes, such as inhibitory pathways or mitochondrial dysfunction. This limitation could significantly impact the accuracy and completeness of SBI predictions, especially for complex pathologies. It would be interesting to know if the authors plan to address these limitations and which future works should be implemented to improve the model and interpretation. For example, by adding parameters that better represent synaptic interactions, inhibitory effects, astrocytic influences.

We completely agree with the reviewer that the accuracy and utility of SBI are inherently tied to the quality of the underlying biophysical model. If crucial biological processes are omitted, the model risks being misspecified, which can lead to incorrect inferences. This is a well-recognized challenge

in SBI, and recent methods have been introduced to assess how likely misspecification may apply. In our revision, we have now explicitly addressed this issue and include a recently published test (Schmitt et al., 2024) to evaluate potential misspecification. Encouragingly, our results indicate that our model is unlikely to be misspecified and likely contains all biological processes relevant to simulate the observed data. Details are provided in a new supplemental Figure 4.

Figure 1: New supplemental figure S4: Check for model misspecification using Maximum Mean Discrepancy (MMD) as proposed by (Schmitt et al., 2024). Shown is the MMD sampling distributions under the training model (H_0), the critical MMD value based on its 95-percentile, and the MMD of all experimental data used (MMD Real vs. Sim).

Regarding the inclusion of inhibitory effects and astrocytes: our current dataset is derived from a co-culture system of excitatory hiPSC-derived neurons and healthy (rodent) astrocytes. Inhibitory neurons are absent from the experimental system and thus not modelled. Similarly, the healthy astrocytes are assumed to be functionally stable in both control and disease conditions and are represented in the model indirectly through fixed parameters. We agree that incorporating patient-derived astrocytes or GABAergic neurons in future experiments could reveal additional disease mechanisms. We are currently collaborating on the integration of hiPSC-derived inhibitory neurons into our cultures and model.

Finally, mitochondrial function is not explicitly modeled but could be included in future work, particularly in disease contexts where it plays a central role. These limitations and future directions are now explicitly discussed in our revision:

*“Employing SBI with our biophysical model presents inherent challenges. SBI relies on the assumption that the computational model sufficiently captures the biological processes underlying the experimental observations. If the model does not accurately reflect the system it aims to describe—known as model misspecification—then even a well-calibrated inference procedure can return misleading parameter estimates²⁹. If the simulator is incomplete, inferred parameters may compensate for unmodeled effects rather than reflecting true biophysical properties. While our model is previously validated with *in vitro* experiments^{10,11}, all models remain simplifications. To assess potential misspecification, we applied the diagnostic test proposed by Schmitt et al.³⁰ and found no clear evidence of model misspecification for our model and experimental data (Supplementary Fig. 4).*

Nevertheless, some disease mechanisms may remain unmodeled. For instance, mitochondrial function is not explicitly included. Moreover, we inferred only 10 model parameters while keeping others fixed, which may limit the approach’s flexibility in capturing diverse disease mechanisms and experimental observations. Expanding the number of inferred parameters would increase computational costs due to the need for more simulations to train the NDE. Additionally, it could complicate the interpretability of the inferred posterior

distributions, due to parameter redundancy and the widening of marginals. While inferring all model parameters and fixing uninformative ones post hoc could aid interpretation, this was not computationally feasible here.

A key limitation of any in vitro-based approach is the gap between the in vitro system and the complexity of the human brain. While in vitro neuronal networks offer a valuable experimental platform for studying disease-related changes in neuronal function, they are much simplified with respect to the in vivo neuronal environment. Therefore, it is essential to exercise caution when extrapolating findings beyond the in vitro system. While our in silico model accurately reflects the in vitro system, it may not capture all mechanisms relevant to pathology in vivo. For example, the in vitro system lacks inhibitory neurons and patient-derived astrocytes. Although our model is not misspecified in this regard (since it accurately reflects the experimental conditions), mechanisms related to inhibitory or astrocyte dysfunction cannot be inferred. However, if inhibitory neurons and patient-derived astrocytes were to be included in future experiments, the model could be extended accordingly to enable inference of inhibition- or astrocyte-related mechanisms^{33,34}.” (Lines 318-348)

1.2. The inability to validate the full posterior distribution limits confidence in the inferred parameters, and in the ability of the method to infer disease mechanisms. Posterior predictive checks are useful, but they do not replace direct experimental validation of inferred mechanisms, which is emphasized but not fully addressed. Include additional experimental evidence to validate the key predictions.

Thank you for these insightful comments. We fully acknowledge that posterior predictive checks alone do not constitute complete validation of inferred parameters. To further assess the reliability of SBI, we now include an analysis of the Parameter Recovery Error (PRE) across multiple parameter combinations. This analysis is added to the Methods and visualized in a new panel in Supplementary Figure 1:

Figure 2: New Panel B of Supplementary Figure 1: Violin plots of the Parameter Recovery Error (PRE) of every parameter for 25 synthetic input data.

That said, we would like to emphasize that the goal of this manuscript is not to draw definitive conclusions about disease mechanisms in specific patient lines. Rather, our objective is to introduce SBI as a novel methodology for analysing MEA data. We investigated whether SBI was able to identify altered cellular mechanisms that were previously found *in vitro* and reported in earlier work. SBI correctly identified nearly all mechanisms reported in the experimental literature, reinforcing confidence in our approach and highlighting the suitability of SBI for application to MEA data.

It nevertheless remains important to highlight and reiterate the importance of experimental validations when researchers apply this method to new MEA data. Therefore, we included in the manuscript:

“We thus strongly recommend performing sufficient experimental replicates and comparing univariate marginals per MEA to identify parameters that are altered consistently. Thirdly, targeted in vitro investigations of SBI-identified disease mechanisms are imperative to validate predictions. In some cases, invalid inferences could result from undetected model misspecification.” (Lines 407-411)

1.3. The lack of reproducibility across batches (e.g., potassium channel conductance in CACNA1A^{+/-} neurons) undermines confidence in some conclusions. Due to the well-known variability of MEA experiments, I recommend performing additional replicates or explore/investigate better technical reasons for variability.

We agree that variability across MEA-batches—such as the observed differences in potassium conductance in CACNA1A^{+/-} neurons—poses challenges for reproducibility. However, this reflects a known limitation of the *in vitro* MEA system, not of SBI or our biophysical model. Our aim is not to optimize the *in vitro* model itself, but rather to demonstrate how SBI can extract mechanistic insights from MEA data, despite experimental variability.

Importantly, this variability and the technical reasons underlying it have been previously addressed by one of our co-authors, dr. M. Frega, in Mossink et al., 2021. They showed that despite the high reproducibility of the differentiation protocol (Frega et al., 2017), variability can be caused by cellular density and distribution, as well as the use of different neuronal and astrocyte batches. They provide experimental guidelines to ensure proper dissection of control and diseased networks. These guidelines were adhered to in our previous experimental studies (Hommersom et al., 2024; Van Hugte et al., 2023), from which we use the data for this manuscript. Thus, additional *in vitro* experiments will likely not further improve the reproducibility, since the remaining variability is intrinsic to the *in vitro* model.

We now better highlight the need for sufficient experimental replicates and the causes of variability in the manuscript:

“Principal-components analysis conducted previously highlighted the influence of these factors on MEA data features², and here we observed their effect on SBI-identified posterior distributions. Secondly, conducting multiple comparisons using independent neuronal preparations on MEA is essential to discern changes consistent across batches. Previous advice emphasized the importance of employing a sufficient number of experimental batches², and here we observed that not all parameter differences were reproduced across batches. We thus strongly recommend performing sufficient experimental replicates and comparing univariate marginals per MEA to identify parameters that are altered consistently.” (Lines 402-409)

1.4. In some paragraphs, the text is sometimes difficult to follow, especially in discussing parameter space and statistical methods. Simplify descriptions where possible and describe SBI keeping in mind a broader audience.

We appreciate this feedback and have revised the manuscript to improve readability, particularly sections discussing parameter space and statistical methods. We simplified technical explanations and clarified key concepts related to SBI to make the text more accessible to a broader neuroscience audience. If there are any specific areas that remain unclear, we would be happy to further refine them.

1.5. Final recommendations: The article is promising but requires significant revision. If the authors can address the validation and reproducibility concerns and improve the clarity of the text, this paper is a good candidate for Nature Communications Biology. The novelty, methodological innovation, and potential impact align well with the journal's scope.

We are very grateful for the reviewer's recognition of the novelty and potential impact of our work. We hope the additional explanations and revisions have addressed the concerns regarding model validity, reproducibility, and clarity.

Reviewer #2:

2.1.1 Summary of Manuscript

In this paper the authors utilize a machine-learning approach named simulation-based inference (SBI) to estimate model parameters of a computational model of human induced pluripotent stem cells (hiPSCs) derived to neuronal networks on multi-electrode arrays. SBI can be used to identify the model parameters that can replicate the neuronal network activity. Here in this case, parameters that were finely tuned based on distributions to set the model parameters to replicate MEA neuronal network activity. Not only was this capable in a physiological normal condition, but SBI can also identify network activity that is sufficiently altered by pharmacologically induced changes, as shown with the case of a network with the addition of Dynasore. The network burst duration decrease was seen in the *in vitro* experiments and modeled in the simulations, and when multiple simulations resultant p-values were averaged the recovery time scale of the short-term synaptic depression was what was consistently concentrated at low values across the models. The paper also showed how if the pharmacological response was not consistent across networks, as the case with the MK-801 NMDA-receptor blocker, then only minor differences are detected. Hence this led the authors to declare that SBI cannot detect molecular changes that do not significantly affect the neuronal network pattern. SBI was also able to replicate disease mechanisms in genome-edited neuronal networks, in this case that for Dravet Syndrome (DS), Generalized Epilepsy with Febrile Seizures (GEFS+), and a case with a monoallelic frameshift variant in exon 8 of CACNA1A. For DS and GEFS+, the SBI was able to, over two batches, detect shifts to lower values in the noise, sodium conductance, and potassium conductance. As well as an increase in short-term depression which was noted to recently in the literature be associated with GEFS+ and DS networks. For the altered CACNA1A, over three batches, the SBI found significant changes toward lower values in the noise, AMPA conductance and connectivity: in one batch a significant shift to lower values in potassium conductance. Overall, this paper sets the framework on how SBI may be used as a method to detect disease mechanisms and alterations in MEA neuronal networks.

2.1.2 Overall Impression of the work

This work is novel in applying the SBI method to MEA networks to get extract the dynamics of the network specifically through the changes in the model parameters. It is an interesting approach to see how altering model parameters can help unravel the changes that are occurring within neuronal network. This paper shows that the process to do is still a bit in its infancy as there are still limitations such as the need for posterior distribution using multiple MEA measurements, reliance on model completeness and computational expense. However regardless of these limitations SBI allows for a robust method to analyze a model's parameters in relation to altered conditions of the neuronal network.

We thank the reviewer for taking the time and effort to review our manuscript, and for recognizing the novelty of our work. Below, we address each of the reviewer's comments in detail and outline the corresponding revisions made to the manuscript.

2.3. Specific comments, with recommendations for addressing each comment

2.3.1. Minor comment, a quick fix for Figure 4 caption C is missing a “ns” before Average P>0.05.

This part of the caption was changed entirely due to a changed figure panel (please see our response to the comment of Reviewer 3: minor comment R4.4).

2.3.2. In lines “93 We analyzed the resulting simulations by computing 13 MEA features that capture the 94 most important characteristics of the network activity and are often used in MEA literature (see 95 Methods and Table 2 for details).” Here perhaps or in the methods

include the literature or justification on why these were the 13 parameters selected and others were excluded (such as how was done with the high correlations between the correlation coefficient between binarized spike trains but include other examples and their respective justifications rather than generalized that too many parameters cause too many variation).

We now better explain why we decided to use hand-crafted MEA features, and we explored an alternative data-driven option (please see reply to Reviewer 3: comment 3.2.2). It is, however, important to note a confusion here regarding the 13 MEA features (start of the reviewers' comment) and the 10 model parameters (end of the reviewers' comment). The latter were indeed limited to 10 because too many parameters might cause too much variation. For the former, however, more features usually result in less variation (please see the new Supplementary Figure 5). The reasons for choosing certain MEA features and for choosing certain model parameters as "free" parameters are quite different. We included a more elaborate explanation for both:

Concerning the MEA features:

"SBI can be trained on hand-crafted summary features, as well as raw time series data, or embeddings learned by neural embedding networks, among others. As our goal is not to replicate the full time series, but rather properties that remain consistent across multiple experimental replicates, we chose the use of summary statistics. While embedding networks can offer a data-driven approach to find informative summary statistics, we opted for predefined MEA features due to their robustness across different recording durations and settings, their widespread use in the MEA field, and their interpretability in relation to known (patho)physiological mechanisms^{2,3,7,35}. This facilitates comparisons across studies and ensures relevance within the MEA research community." (Lines 353-361)

Concerning the free model parameters:

"To ensure computational feasibility and model interpretability, we fixed specific model parameters related to membrane and synaptic time constants while allowing others, such as ion channel conductances and synaptic strengths, to vary. The membrane time constant emerges from passive properties such as the membrane capacitance and leak conductance, both of which remain stable in most disease conditions⁴⁶. AMPAr and NMDAr decay are fixed because they are determined by receptor subunit composition, which is developmentally regulated and remains stable under most pathological conditions⁴⁷. Ion channel time constants are highly correlated with the expression of that ion channel¹⁴, so allowing both time constants and maximal conductances to vary would result in parameter redundancy and less interpretable posteriors. Therefore, we fixed time constants while keeping ion channel conductances free to vary. Lastly, Nernst potentials were fixed because they are mainly governed by astrocyte function. Since our cultures have healthy astrocytes, not patient-specific ones, we assume the astrocyte-related contributions to remain constant and unrelated to disease mechanisms." (Lines 472-484)

2.3.3. One comment In lines "373 it is crucial to compare neuronal networks cultured on the same MEA and measured simultaneously, as differences in astrocyte and MEA batches can introduce confounding factors." The computational model and study include no mention of astrocyte modeling and therefore it may be important to make note of this and how it may still be unknown if SBI would be able to test for mechanisms including astrocytes as this analysis was done primarily on a neuronal network model. It may be important to note this that this analysis of certain parameters and their dynamics do depend on the model and its parameters that are identified.

Indeed, we did not explicitly model astrocytes, and therefore, SBI cannot identify disease mechanisms that include astrocytes. This is an inherent limitation of our *in vitro* system, which includes healthy (rodent) astrocytes that are assumed to function stably and consistently across

experimental conditions, since they are not patient-derived. We model their effects indirectly, assuming stable reversal potentials and neurotransmitter reuptake. However, indeed, if one would like to research mechanisms that include the astrocyte, patient-hiPSC-derived astrocytes can be included *in vitro*. Then, astrocytes can also be explicitly incorporated in our *in silico* model, using e.g., the model by Lenk et al. (Lenk et al., 2020). We now write:

*“While our *in silico* model accurately reflects the *in vitro* system, it may not capture all mechanisms relevant to pathology *in vivo*. For example, the *in vitro* system lacks inhibitory neurons and patient-derived astrocytes. Although our model is not misspecified in this regard (since it accurately reflects the experimental conditions), mechanisms related to inhibitory or astrocyte dysfunction cannot be inferred. However, if inhibitory neurons and patient-derived astrocytes were to be included in future experiments, the model could be extended accordingly to enable inference of inhibition- or astrocyte-related mechanisms^{33,34}.” (Lines 339-348)*

The reviewer is also correct to notice that the analysis depends on the model and the parameters that we allow to be identified. We now more explicitly discuss this in the manuscript:

*“SBI relies on the assumption that the computational model sufficiently captures the biological processes underlying the experimental observations. If the model does not accurately reflect the system it aims to describe—known as model misspecification—then even a well-calibrated inference procedure can return misleading parameter estimates²⁹. If the simulator is incomplete, inferred parameters may compensate for unmodeled effects rather than reflecting true biophysical properties. While our model is previously validated with *in vitro* experiments^{10,11}, all models remain simplifications. To assess potential misspecification, we applied the diagnostic test proposed by Schmitt et al.³⁰ and found no clear evidence of model misspecification for our model and experimental data (Supplementary Fig. 4)*

Nevertheless, some disease mechanisms may remain unmodeled. For instance, mitochondrial function is not explicitly included. Moreover, we inferred only 10 model parameters while keeping others fixed, which may limit the approach’s flexibility in capturing diverse disease mechanisms and experimental observations.” (Lines 318-332)

Reviewer #3:

Disclaimer: I am not familiar with hiPSC-NNs data, but have practical experience in the analysis of in-vivo and in-vitro mammalian electrophysiological signals. Where I may have misunderstood the authors' intent, I encourage them to point it out in their responses, and if applicable, to kindly understand an implicit need for a clearer explanation to makes the work accessible to a wide audience of neuroscientists from different subfields.

Overall impression

****The authors estimate the parameter values that allow a biophysical model of excitatory neuronal networks to closely reproduce multielectrode recordings of the activity of a network of cultured** **hiPSCs. They claim that their approach using simulation-based inference yields estimates that plausibly track pharmacological and gene-editing interventions and even allows to identify key disease mechanisms.****

The present study extends to hiPSC-NNs the recent trend of using SBI for reliable inference of inverse problems in neuroscience, which in my view is both timely and relevant, given the claimed diagnostic potential of hiPSC-NNs. The text is well articulated (in particular the introduction does an admirable job of presenting the relevance of the problem, typical approaches, shortcomings and the advantages of SBI with scant use of machine-learning jargon), the schematics and data displays are well organized and clearly presented, and the results are critically discussed with transparency about the limitations of the approach.

With some technical improvements to buttress the authors' use of sbi, with which I have chiefly concerned myself for this review, I believe this will become an influential study, since it exemplifies how to interrogate hiPSCs-NNs with state of the art inference techniques and presents findings in a balanced and accesible way to neuroscientists at large. Even if the SBI approach "just" accelerates hiPSCs-NNs fingerprinting by providing reliable parameter estimates otherwise only accessible via direct experiments, this constitutes a very significant speed up for research both on neuronal network as complex systems and of their spontaneous activity as a subtle biomarker for clinical conditions.

We would like to express our sincerest thanks to the reviewer for his exceptionally thorough and constructive review. The level of detail, constructive suggestions, and insightful discussion demonstrates a remarkable commitment to both scientific rigor and the advancement of our work. We sincerely appreciate the time and effort invested in analyzing our manuscript, providing alternative perspectives, and guiding us toward meaningful improvements. The review not only helped us strengthen our manuscript but also provided us with valuable insights that will benefit our future research. It is furthermore meaningful to us that the reviewer recognizes the potential and added value of our approach, and we appreciate this encouragement.

Major comments

3.1. **Research design and scope of conclusions.** While pharmacological and gene-editing interventions can be performed directly on object of the simulation, namely the cultured tissue, a disease affects the organism from which the hiPSCs are sourced. Cultures of a diseased individual's hiPSCs-NNs act as a proxy for a much more complex neural system; it is crucial to exercise caution (and warn the reader) about drawing conclusions for living human brains from parameter values obtained with SBI (indirection #1) based on comparing simulated electrical activity (indirection #2) with experimental multielectrode recordings (indirection #3) of in-vitro tissue (indirection #4). As long as the cultured tissue and the simulator are only used to fingerprint activity based upon which SBI can flag potential conditions for further

investigation in vivo, that is safe territory. This limitation is shared with similar research, so it is only a matter of finely acknowledging where appropriate, for example:

We agree with the reviewer. We do not intend to suggest that *in vitro* disease mechanisms are directly translatable to the *in vivo* situation. Indeed, the point of our method is to automatically identify candidate mechanisms that may underlie observed differences in the *in vitro* electrical activity. These findings cannot be directly extrapolated to disease mechanisms in the human brain, but can give directions for further research. We have made this point more explicit throughout the work, including wording to caution against overinterpreting *in vitro* findings.

3.1.1. clarifying in L316-319 that the simulator model is designed for a closed (in complexity and spatial extent) cultured system, and that statements pertaining to disease, i.e. about the open-system, in-vivo network, are additionally uncertain, reminding that that both the simulator and the direct measurements with which parameter inferences are compared refer to cultured tissue

We rewrote this part of the discussion and split it into a paragraph about model misspecification and one about caution when extrapolating findings to the *in vivo* system. We clarified the latter as follows:

“A key limitation of any in vitro-based approach is the gap between the in vitro system and the complexity of the human brain. While in vitro neuronal networks offer a valuable experimental platform for studying disease-related changes in neuronal function, they are much simplified with respect to the in vivo neuronal environment. Therefore, it is essential to exercise caution when extrapolating findings beyond the in vitro system. While our in silico model accurately reflects the in vitro system, it may not capture all mechanisms relevant to pathology in vivo. For example, the in vitro system lacks inhibitory neurons and patient-derived astrocytes. Although our model is not misspecified in this regard (since it accurately reflects the experimental conditions), mechanisms related to inhibitory or astrocyte dysfunction cannot be inferred. However, if inhibitory neurons and patient-derived astrocytes were to be included in future experiments, the model could be extended accordingly to enable inference of inhibition- or astrocyte-related mechanisms^{33,34}” (Lines 338-348)

3.1.2. while L19 in the abstract is clear in this respect, I am uncertain about readers expecting too much in terms of elucidation of in-vivo mechanisms based off L10-L13 and L20, L22. I recommend to explicitly separate the statements on disease mechanisms from the others and use more cautious language

We believe L10-13 is clear in this regard because there is a specific reference to pathology underlying *in vitro* neuronal network, not *in vivo* pathology. We made this more specific in Line 20 and changed:

“This demonstrates SBI’s potential to automate and enhance the discovery of in vitro disease mechanisms from MEA measurements,” (Line 20).

3.1.3. L34 furnishes, in my view, an example of adequate language, with L35-36 situating the work as part of in-vitro based hypothesis making about in-vivo systems, which the paper intends to substantially accelerate

3.1.4. see Minor comments > Discussion > Limitations > 1 for more specific suggestions

3.2. **Limitations common simulator-based research. The following limitations are common to research based on simulators, and as such they do not challenge this study specifically but rather as part of an entire approach to researching complex systems. In fact, this study should be commended for adopting SBI and thus entirely avoiding a third common limitation: the inability of most popular scalable inference techniques to identify multiple parameter sets compatible with the observations and provide meaningful error estimates.**

Thank you, and we agree.

3.2.1. The simulator may not adequately represent the system (“model misspecification”). The target here should be to ensure the simulator is well specified wrt. the in-vitro system; the complexity of the in-vivo tissue would likely be out of the scope of present SBI techniques in terms of number of parameters involved. Since the literature on diagnostics for model misspecification in the SBI field is stil nascent [1, 2, 3], and the authors use a previously published and validated forward model for hiPSCs-NNs on a best-effort basis, I recommend, that authors, at their discretion and for increased trust,

- 1. run checks proposed in [1],**
- 2. or, if using embedding networks (see below and in minor comments to Methods), that they use those proposed in [2]. This might represent a considerable effort (albeit less than for those suggested in [1]), but, in any case,**
- 3. authors should refer to model misspecification by name and outline what steps appropriate to their system they or others could undertake in future work to diagnose potential misspecification**

Indeed, model misspecification is a common pitfall in research based on simulators. In previous research, we validated our model experimentally by comparing responses to *in vitro* and *in silico* perturbations (Doom et al., 2023, 2024a). Moreover, we showed that, irrespective of whether the model is perfectly specified or not, it was able to generate predictions that were experimentally confirmed. We consider this predictive capacity a crucial aspect of our application. However, we had not yet performed any of the mentioned diagnostic tests for model misspecification. We thank the reviewer for these suggestions and have now evaluated our model accordingly.

We now performed the test proposed in [2], which, according to the paper, can also be applied to hand-crafted summary statistics. We normalized and whitened all summary statistics and computed the maximum mean discrepancy (MMD) between all experimental data used in this study and the simulations used for training, using an inverse multiquadratic kernel with $h=0.1$ as suggested in (Ardizzone et al., 2019). We determined the critical MMD value by estimating the sampling distribution under the null hypothesis from multiple sets of simulations from our model, as outlined in [2]. This sampling distribution, the critical MMD value, and the MMD value of our experimental data are shown in the new Supplemental Figure 4. The MMD values we computed were surprisingly low compared to those reported in [2]. This might be due to our use of 15-dimensional, hand-crafted, normalized summary statistics, whereas [2] typically considers lower-dimensional embeddings generated via neural networks. Nevertheless, this analysis suggests that our model is not misspecified. The code to reproduce this analysis is available in the project repository.

Figure 3: New supplemental figure S4: Check for model misspecification using Maximum Mean Discrepancy (MMD) as proposed by (Schmitt et al., 2024). Shown is the MMD sampling distributions under the training model (H_0), the critical MMD value based on its 95-percentile, and the MMD of all experimental data used (MMD Real vs. Sim).

We also explicitly address the risk of misspecification in our revised manuscript:

“Employing SBI with our biophysical model presents inherent challenges. SBI relies on the assumption that the computational model sufficiently captures the biological processes underlying the experimental observations. If the model does not accurately reflect the system it aims to describe—known as model misspecification—then even a well-calibrated inference procedure can return misleading parameter estimates²⁹. If the simulator is incomplete, inferred parameters may compensate for unmodeled effects rather than reflecting true biophysical properties. While our model is previously validated with in vitro experiments^{10,11}, all models remain simplifications. To assess potential misspecification, we applied the diagnostic test proposed by Schmitt et al.³⁰ and found no clear evidence of model misspecification for our model and experimental data (Supplementary Fig. 4).

Nevertheless, some disease mechanisms may remain unmodeled. For instance, mitochondrial function is not explicitly included.” (Lines 318-330)

3.2.2. The parameter inference in SBI might be based on a reduced set features of the measurements due to investigative focus, interpretability concerns or computational limitations, so that inference may be underconstrained or not attend to key dynamical aspects of the system (insufficiency of “summary statistics”). In this study, the authors reduce one-minute time series to 13 time and frequency-domain scalars (e.g. rates and inter-event intervals of spikes and bursts), but it is unclear if these can fully capture the underlying dynamics (cf. cautionary example in [4]), even if the resimulated traces reproduce the bursting behaviour relatively well. To strengthen their study, I recommend to compare current results from hand-crafted features with those derived from employing either

- 1. an unsupervised technique for dimensionality reduction, such as PCA (as demonstrated in [4]), or, better yet,**
- 2. a data-driven approach to summary statistics, by using an embedding network (something that is enabled by the author's choice of using neural posterior estimation, NPE). Relevant examples of this approach are [5, 6]; both using convolutional networks and some form of transfer learning and the second specific to neuroscience time series. See minor comments to the Methods for more concrete recommendations.**
- 3. Regardless of the path taken, authors can compare results as outlined in [4, Fig 4] with PRE (parameter recovery errors) besides PPC; this reference also proposes OVL as a**

means to compare posterior distributions; see also [11] for feature importances based on KL distances of posteriors.

We thank the reviewer for raising this important point regarding the potential insufficiency of summary statistics and the possible benefits of using data-driven alternatives.

We now realize that we did not explicitly mention in the manuscript that the 13 MEA features are not computed from one-minute time series, but rather from longer recordings—either 3-minute simulated traces or 5- to 10-minute experimental recordings. We have now clarified this in the Methods section:

“We sampled 300,000 parameter configurations from the prior distribution (a uniform distribution within the ranges shown in Table 1) and performed 3-minute simulations with the computational model with each of these configurations.” (Line 539)

Our decision to use hand-crafted summary statistics is motivated by their robustness to variations in recording length and experimental conditions. Most recordings are 10 minutes long and sampled at 10 kHz, but some are shorter and sampled at 12.5 kHz. In contrast, simulations are only 3 minutes long due to computational constraints. Our summary features can be consistently computed across these conditions, ensuring comparability between experimental and simulated data and maximal use of the full recordings.

Moreover, these features are well-established in the MEA field and have known associations with neurobiological processes (Frega et al., 2019; Maheswaranathan et al., 2012; Mossink et al., 2021; Van Hugte et al., 2023). Their familiarity also facilitates interpretability and broader adoption within the neuroscience community.

That said, we fully agree with the reviewer that our hand-crafted features may not be maximally informative and that data-driven features might improve inference. We also acknowledge the reviewers' comments on the Methods regarding the ad-hoc burst detection settings, which were initially based on expert visual inspection.

To assess whether inference could benefit from data-driven features, we implemented an alternative approach using an embedding network. We opted not to use PCA as it does not capture temporal dynamics when applied to our multi-unit recordings or spatial dynamics when applied to e.g. array-wide firing rates.

We designed and trained a 2D-convolutional network to capture both spatial and temporal structure. We could not find any existing such network for application to multi-unit recordings. Since we did not save the raw simulated activity of the 300,000 simulations used in the manuscript (only the computed features), we now simulated an additional 100,000 configurations (saving spike times) and used these to train the embedding network. Since the simulated and experimental voltage traces are at least 3 minutes long and sampled at 10 kHz, we instead used firing rates (spikes gathered in bins) to limit the dimensionality of the input to the embedding network. This approach thus still requires an arbitrary choice for bin size, and even after binning with a reasonable bin size (100 ms), the input remains high dimensional (12×1800). We designed an embedding network architecture to appropriately handle this high-dimensional input and extract features at appropriate temporal scales (bursting-like features at a large scale and spiking activity at a small scale). The architecture included:

1. Local spatio-temporal feature extraction: Conv2D with (3,5) kernel size (electrodes \times time)
2. Temporal feature extraction: A depthwise convolution with (1,5) kernel.
3. Electrode interaction encoding: A (1 \times 1) convolution to integrate information across electrodes.

4. Capture Long-range temporal dependencies: Two dilated convolutions with (1,5) kernels and increasing dilation rates (2 and 4).
5. Dimensionality Reduction: To control feature map growth (we ran into the limits of our RAM), we used two max-pooling layers with kernel size (1,2).
6. Flattening, a fully connected layer (256 neurons, ReLU activation), and a final linear output layer that maps to a 20-dimensional embedding space.

We trained the embedding network jointly with the NDE. We also trained a separate NDE on our hand-crafted summary features extracted from the same 100k simulations, to compare to performance of inference with the embedding network. Joint training of the embedding network and NDE took $\sim 100x$ longer than using hand-crafted features, and inference time was similarly increased. Therefore, we were not able to extensively tune the embedding network architecture.

Unfortunately, the performance of inference with the jointly trained networks on synthetic data was poor. With the embedding network, the marginals of the posterior distribution were very wide, the PRE was high, and performance was inferior to our original approach. A representative example is illustrated in Figure 4.

Figure 4: Posterior distributions inferred on synthetic data using an NDE trained on A) hand-crafted MEA features and B) the output of a 2D-convolutional embedding network. C) Middle: 1.5-minute rasterplot showing the simulation on which inference was carried out; Left: simulation with the mode of the posterior shown in panel A; Right: simulation with the mode of the posterior shown in panel B.

The average PRE with hand-crafted MEA features was 0.0728 compared to 0.1670 with the embedding network. We think that the poorer performance of the embedding network may stem from our unoptimized architecture or the constraints on the size of the embedding network due to our limited computational resources. However, considering the time it took to train and perform inference with this set-up, and arguments explained above, we argue hand-crafted, community-vetted summary features are most appropriate for our application. Nevertheless, we cannot rule out the

potential of embedding networks given better resources. The code to reproduce these results is available in the repository.

As suggested by the reviewer, we now also computed the Parameter Recovery Error (PRE) for each of our configurations used for posterior-predictive checks. This is included in the methods:

“Additionally, we computed the Parameter Recovery Error (PRE) for each parameter for each of the twenty-five configurations, which is a measure of how concentrated the marginal is around the ground truth parameter value⁵⁰” (Lines 558-561)

We added a new panel to the supplemental Figure 1 (Referred to in Results line 108):

Figure 5: New panel B of supplementary Figure 1: Violin plots of the Parameter Recovery Error (PRE) of every parameter for 25 synthetic input data.

In the reference [4] cited by the reviewer, PRE is computed over a 2D grid of parameter values to gain insights into regions of poor inference performance. However, in their work, they only infer 3 parameters. In contrast, our model infers ten parameters, making a grid-based exploration computationally challenging and difficult to visualize. For these reasons, we opted to report overall PRE values per parameter, which still provides a useful measure of inference accuracy.

3.2.3. ****Limitations in this study's use of the SBI approach, and its reporting for review and reproducibility efforts.****

3.2.3.1. Authors claim to use a mixture-density network (MDN) as posterior density estimator, but the published code in the accompanying repository at https://gitlab.utwente.nl/m7706783/SBI_MEA_modelis quite incomplete, so that the interested reader cannot find how the MDN is parameterized, or, in fact, any of the training code at all. This is a severe shortcoming and for me as a reviewer who desires to understand the work, a red flag for publication (a false appearance of transparency is worse than the lack of transparency). I recommend that in a further submission:

- 1. The accompanying code should be complete. Publishing just the network weights, despite PR attempts by e.g. Meta with their Llama models, falls short of fully open sourcing the model, which is necessary for proper reproducibility and hence for scientific discourse. Weights are helpful, but the full training protocol is essential and, together with the measured data and simulation results (again, better the simulator source code), allows any interested party to recover the weights just with some computational effort.**
- 2. The density estimator should also be briefly described in the Methods section of the main text, indicating number of layers, GMM components, etc,**

- 3. The peculiar choice of an MDN should either be quantitatively justified, or a more powerful density estimator should be adopted (see e.g. the seminal publication [16], cited by the authors). For context, I surveyed related applied literature wrt. choices of density estimators and found 10 references ([4, 5, 6 7, 8, 10, 12, 13, 15, 16]) preferring normalizing flows, most of the time a Masked Autoregressive Flow (MAF; [the default for NPE in the sbi toolkit](https://github.com/sbi-dev/sbi/blob/f2c1cc3d623f8a8a71eb3c132357b2d376314b73/sbi/inference/trainers/npe/npe_base.py#L51)) and occasionally a Neural Spline Flow (NSF; performing better than MAF in the SBI benchmarking paper [18, Appendix H.5]), vs. a single reference ([9]) adopting a MDN.

We sincerely apologize for the confusion regarding our choice of neural density estimator. This was an unfortunate oversight on our part. During the early stages of our project, we explored multiple density estimators, including Mixture Density Networks (MDNs). However, in our final implementation, we used the default setting for NPE in the *sbi* toolkit, which is indeed a Masked Autoregressive Flow (MAF). Unfortunately, we made a mistake in reporting our methods, and incorrectly kept our use of an MDN in the text. We apologize again for our negligence and for overlooking this mistake before submitting our manuscript.

To address this, we have now corrected the Methods section to accurately reflect the use of MAF:

*“This NDE was the standard **Masked Autoregressive Flow** provided by the SBI toolbox.”*
(Line 542)

Regarding the completeness of our code, we initially assumed that providing the training dataset would be sufficient for reproducibility, given that we used the default *sbi* settings. However, for completeness, we have included the complete training protocol in our repository to ensure that all results can be independently reproduced.

3.2.3.2. When conducting inference, authors decide to leave some parameters fixed, adducing that the interpretation would be further complicated by the washed-out marginals they expect in case of higher parameter space dimensionality. I believe these improvements could be considered:

1. **Provide model-specific, neuroscientific rationale for why the authors decided to consider some specific parameters as fixed**
2. **(effortful) If the computational performance of the potential new density estimator allows, leave more (ideally all) now-fixed parameters free for adjustment within sensible prior ranges; After inference, those parameters for which the range of variation is (fractionally) small can be fixed for discussion and interpretation. That is, authors could already in Fig. 1B use conditional posteriors, where those parameters that have “uninteresting” ranges act as conditioners. This makes sense insofar that it allows SBI to find potentially better combinations of parameters in the enlarged space, yet authors can focus the discussion on (potentially) the very same parameters they focus on today (or others, depending on the outcome).**

We thank the reviewer for the suggestion of leaving all parameters free and using conditional posteriors for discussion and interpretation. This might be the ideal case for our application. However, unfortunately, we do not have the computational capacity to perform enough simulations to train the neural density estimator for so many free parameters. The gathering of the current simulation dataset took multiple months, and we suspect that inference of all (around 35) free parameters would require many more than 300,000 simulations, likely well beyond our current capacity.

We did now provide model-specific, neuroscientific rationale for why we decided to consider specific parameter fixed:

“To ensure computational feasibility and model interpretability, we fixed specific model parameters related to membrane and synaptic time constants while allowing others, such as ion channel conductances and synaptic strengths, to vary. The membrane time constant emerges from passive properties such as the membrane capacitance and leak conductance, both of which remain stable in most disease conditions⁴⁶. AMPAR and NMDAR decay are fixed because they are determined by receptor subunit composition, which is developmentally regulated and remains stable under most pathological conditions⁴⁷. Ion channel time constants are highly correlated with the expression of that ion channel¹⁴, so allowing both time constants and maximal conductances to vary would result in parameter redundancy and less interpretable posteriors. Therefore, we fixed time constants while keeping ion channel conductances free to vary. Lastly, Nernst potentials were fixed because they are mainly governed by astrocyte function. Since our cultures have healthy astrocytes, not patient-specific ones, we assume the astrocyte-related contributions to remain constant and unrelated to disease mechanisms.” (Lines 472-484)

Minor comments

Introduction

- - I1. L45 and the network CONNECTIVITY (looking e.g. at Table 1, it seems that the network is represented here just by the synapses and the wiring probability and spatial patterning)
 - I2. L51 with -> using, as, via
 - I3. L59 one could add in parenthesis, to better convey how damning and inescapable this phenomenon is, the term of art in machine learning: “curse of dimensionality”
 - I4. L60 there is a chance to introduce the concept of amortization, e.g. “(i.e. they are non-amortized)”
 - I5. Ref. 18 is not immediately applicable to the SBI techniques in this paper and can confuse the reader looking for more detail. Ref. 19 is more relevant (using normalizing flows), but should be replaced by a different one (e.g. [17], using MDNs) if the authors present good reasons to keep using the MDN.
 - I6. L69 not using simulations “only”; since SBI is a Bayesian framework, priors provide a structured way to leverage existing knowledge about the parameters, and help speed up inference; the authors likely intend “only” to convey that only sampling, and no evaluation of a likelihood function is needed. Unfortunately, there is no preceding discussion of likelihood-based methods, so this implicit opposition may be confusing for readers as it was for me.
 - I7. L80 does not mention the gene-editing data I8. L75-85 mix tenses

We corrected the text exactly as the reviewer suggested.

Results

SBI correctly identifies...

R1. L91-92

R1.1. “*i.e.*” seems to imply that a (any) prior distribution *is* a uniform distribution. I advise to give due treatment to the idea of a posterior in the introduction (see above) or in a separate sentence, where authors can justify the use of their chosen prior (in absence of

specific reasons, as a customary approach often taken when approaching Bayesian estimation with limited information, i.e. “this parameter must be positive” already fixes one side of the interval, and some physical / physiological reasoning easily fixes the highest allowable value of the parameter”).

R1.2. the term “*box prior”,* used in the sbi library, conveys perhaps better the situation when the uniform distribution over a range is, in fact, multidimensional, i.e. uniform over the cartesian product of several 1D intervals.

R1.3. Fig 1 panel A1 confusingly depicts a prior that is *not* a box prior as discussed above. Since panel A2 introduces details of the actual simulator (instead of remaining at the schematic level as a “black box”), A1 should be at least not contradictory wrt. the text and depict a square of equal probability. Adding “(box prior)” or “(uniform)” to the panel title could support the connection between text and diagram. Similarly for the caption: since in 2 simulations are performed using *our biophysical model*, in 1 the parameters should be drawn from e.g. “within plausible ranges (see Table 1)”

The prior knowledge is now included in the introduction:

“SBI is a machine-learning approach that allows efficient statistical inference of biophysical model parameters using simulations and any prior knowledge or constraints on parameters.” (Line 68).

We refer to a box prior in the Results, Figure 1, and caption of Figure:

“To do so, we sampled 300,000 model parameter configurations from a box prior (Fig. 1A1) with plausible parameter ranges (see Methods and Table 1 for details on parameter choices) and used them for simulations with our biophysical computational model (Fig. 1A2).” (Lines 89-92)

Figure 6: New Panel A of Figure 1 showing a Box Prior. New caption: 300,000 parameter configurations are drawn from within plausible ranges (see Table 1);

R2. L92

R2.1. double parenthesis; use citep or reorder the sentence to facilitate reading (ideally, do not present prior in a parenthesis, since it is an important notion)

Done; see above.

R2.2. no mention is made at this point of why some parameters are considered subject of investigation while others are fixed with, effectively, infinite precision; authors can defer explicitly to the discussion.

Included reference to parameter choice rational; see above.

R2.3. L94 - recommend introducing the term of art: summary features/summary statistics: “13 MEA features that capture the most important characteristics of the network activity (summary statistics)”. The superlative “most important characteristics” is not warranted in a parameter inference context, and I would contest that even from a signal analysis point of view these particular features are the most important. Which are most important can only be decided in a data and task-dependent manner (see discussion of the Methods section below). This remark doesn’t negate the commendable effort of the authors in using many features that cover both time and spectral domain aspects of the recording and are community-vetted

Added “summary statistics” and removed “the most”:

“We analyzed the resulting simulations by computing 13 MEA features that capture important characteristics of the network activity and are often used in MEA literature (summary statistics)” (Lines 92-94).

R2.4. L99-100 - compatible THE PRIORS and experimental observations. Further, a distribution should not be called a space, especially one with infinite tails, as, effectively, every parameter combination would be “compatible with”, albeit most of them not “probable under”, the observations. The posterior distribution has been already introduced in L71; the concept can be directly used

Adapted:

“This resulted in a posterior distribution per measurement, which represents the probability of different model parameters given both the prior distribution and the experimental observations (Fig. 1A6). This distribution assigns high probability (yellow) to parameter values that are most consistent with the input measurement, while regions with low probability (dark blue) correspond to parameters that produce simulations that mimic the observation less well (Fig. 1A7)” (Lines 98-103)

R2.5. L102 - simulations mimicking data? (I take data here to be measurements)

See above.

R2.6. Figure 1A6, caption: the -> one, per -> for each. For clarity, authors should stress early on that a recording is conceptualized as a single observation, and/or be careful when using “observations” in plural, since it may cause confusion: yes, they are observationS in time, but SBI sees them as a single observation, a 13-component vector.

R2.7. Figure 1A6, caption: mixes tenses wrt. 1A1-1A5 and 1A7

R2.8. Figure 1, caption C: separate words in rasterplot (as line plot or bar plot, though admittedly, scatter plot is used also in its single word variant)

R2.9. Figure 1, caption D: please set $n=10$ in math mode for proper spacing, here and elsewhere in the manuscript

Response to R2.6-R2,9:

Caption of Figure 1 was changed to:

“ 6) this results in one posterior distribution for each experimental observation, which are compared to identify mechanistic differences; 7) simulations with parameters sampled from the posterior distribution are similar when compared to experimental observations. ... C) Raster plots showing 1 minute of (left) simulation used as input for the inference, (top right) example simulation with the mode of the posterior distribution, and (bottom right) example simulation with the low probability model parameters. D) MEA features of simulations ($n = 10$ per condition) with the ground-truth model parameters (brown), the mode of the posterior (pink), and low probability model parameters (beige).” (Page 6)

“Raster plot” was separated throughout the manuscript, and $n = 10$ was set in math mode throughout.

R2.10. L107 - use m-dashes for the parenthetical expression (<https://www.merriam-webster.com/grammar/em-dash-en-dash-how-to-use>), and do not write a dash between ground and truth, as this would join those words into an adjective

We have changed this throughout. However, when “ground-truth parameters” is used, we do use the dash because “ground-truth” then functions as an adjective to “parameters”.

R2.11. Fig 2C - is one minute enough to estimate low-uncertainty summary features of MEA activity, in particular those involving ISIs of low-firing cells? This really depends on the observed rates. Note one advantage of summary features is that they allow, in principle, to use different-length recordings (irregular data)

Indeed, as explained above, we always use 3-minute simulations to train the neural density estimator, and our experimental data is at least 5 minutes in duration. We evaluated the duration of simulations needed for stable summary features (not reported) and chose 3 minutes based on that evaluation.

R2.12. Fig 2A, caption: \$p\$-value or P value? I prefer downcase \$p\$-value, as we're talking about a realization, not a random variable (unless Nat Comm Bio stipulates otherwise in their instructions to authors). See discussion at

<https://stats.stackexchange.com/questions/871/correct-spelling-capitalization-italicization-hyphenation-of-p-value>

Changed to \$p\$ throughout the manuscript

R2.13. Fig 2A, and all subsequent figures displaying single values as dots---please use a smaller dot size and jitter the dots horizontally (see e.g. seaborn's stripplot or swarmplot functions). Otherwise, dots occlude each other, e.g. in the bar plot #Fragments/NB there are eight dots over each bar, which means that simulations do not show almost any variability in this metric

All bar plots now have scatter plots on top of them, for example in Figure 2:

Figure 7: New scatter plots on top of bar plots in Manuscript Figures 2,3 and 4, and Supplementary Figure 2.

R2.14. Fig 2A, bar plots, left bar dots - even within a single subject, the variability of summary features seems very substantial . This makes me wonder about using a single control subject for comparison with diseased individuals. Comparisons are made always on inferred parameters - but how distant are the 13 summary features themselves? I would welcome insights from the authors on this issue

Indeed, the variability within a single subject is quite substantial, a known challenge in the field. However, note that simulations with the mode of the posterior distribution (one single set of parameters, the same for each simulation) also produce highly variable results. This suggests that, even though biophysical properties are the same in each network, slight variations in the

random connectivity or neuronal excitability can cause large differences in the observed network activity.

We agree that, ideally, multiple control lines should be included to account for inter-subject variability unrelated to disease. Note that this is not necessary for the genome-edited lines, as there is only one subject. In the experimental study on the patient lines used here, there was an additional control subject included (Van Hugte et al., 2023). However, we noted that to maximize the distinctive capability of SBI, we had to compare networks on the same MEA plate and repeat this comparison for multiple plates (as we also recommend in the manuscript). It is technically challenging to culture multiple control and multiple patient lines on the same MEA plate with a sufficient number of replicates (the most used multi-well MEA is composed by 24 wells), and thus usually only one control is plated together with the patient lines on a MEA. This was also the case for the experimental data we used.

Despite the considerable variability, the MEA summary features are usually quite distant on average. In most studies using hiPSCs on MEA to study disease, linear discriminant analysis or some clustering analysis is used to show that summary features are quite distant when taken all together (Hommersom et al., 2024; Van Hugte et al., 2023).

Here, to investigate how distant our MEA features are between the different conditions we compare, we compute the Energy Distance. We chose Energy Distance because it works in multiple dimensions and handles small samples better than histogram-based methods. The energy distances between standardized MEA features of which we compared the corresponding posterior distributions are as follows:

Comparison	Posterior Figure	Energy Distance
Healthy 1 vs. Healthy 2	Figure 2B vs. Supp Figure 2B	5.25
Before vs After Dynasore	Figure 3A	3.64
Before vs After MK801	Figure 3E	0.71
Control vs DS	Figure 4A	6.04
Control vs GEFS+	Figure 4A	2.44
Control vs CACNA1A ^{+/-}	Figure 4D	1.31

We see substantial differences in activity between conditions, especially in the DS and GEFS+ comparisons. However, the large distance (5.25) observed between two control cultures from the same donor—but recorded years apart on different MEA plates and using different astrocyte batches—underscores the considerable contribution of technical and batch effects

While differences between batches are large, the differences between healthy and diseased lines are often consistent across batches. This is, for example, seen in the DS and GEFS+ lines, where the NBR of the healthy control networks has a two-fold difference between the two batches, but the NBR is significantly lower in DS compared to the corresponding control in all cases.

SBI can identify the entire landscape ...

R3.1. L114 “parameters” cannot be a subject to the verb simulate---they can enable somebody to simulate

Changed to:

“Next, we used SBI to estimate the in silico model parameters that allow us to simulate network activity...” (Line 115)

R3.2. L119 to check/EVALUATE reproducibility (no THE)

Changed

R3.3. L122 the batchwise variability for a single individual makes me wonder whether the variability across individuals is not so large that the notion of a “control” subject loses its meaning

See comments to R2.14

R3.4. L136 and L148 I advise to use not weak resp. high correlations but weak resp. strong associations, since the eye will perceive associations beyond just linear (see comments on Methods about quantification)

R3.5. L141 the remaining ONE or TWO model parameters could take to still REproduce the desired network activity. This resulted in conditional posterior distributionS (since there are distributions conditioned both to k-1 and k-2 parameters)

R3.6. L145 vary within a limited range __ and strongly constrain each other __ in order for simulations to match observations

R3.7. L149 with/OVER many possible CONDITIONING configurations/SETS

R3.8. L151-152 I find the structure of the sentence confusing. Presumably what is meant by the number of synapses is the parameter Conn%. It would be good to reuse the terminology given in Table 1 (probability of connection), even if the number of synapses is related (I am also unsure if the model contemplates multisynaptic connections). Furthermore, the two comparisons should be separate: even if both the AMPA and NMDA conductances were strongly negatively correlated to the probability of connection

Response to R3.4-R3.8; we changed:

*“By examining the pairwise marginals, it seems that parameters are only weakly **associated**, as indicated by large clouds without a clear direction (Fig. 2B). ... To explore this further, we held all but one or two parameters constant at values sampled from the posterior distribution and observed what values the remaining **one or two** model parameters could take to still reproduce the desired network activity. This resulted in conditional posterior **distributions** (Fig. 2C, Supplementary Fig. 2C). We found that these conditional distributions were significantly narrower, confirming that if some model parameters have a certain value, the remaining model parameters can only vary within a limited range—**and strongly constrain each other**—to still yield simulations that match the experimental observation. When looking at remaining possibilities with all but two parameters set to a constant value (Fig. 2C), we observed high **associations** between some model parameters. When generating the conditional distribution **over many possible** sets, we noticed that some conditional correlations were preserved, leading to an average correlation coefficient significantly different from zero (Fig. 2D,E). We found, for example, **a strong negative correlation between AMPA conductance and the probability of connection (Conn%), as well as between NMDA conductance and Conn%. This indicates that fewer synapses can be compensated by stronger synapses and vice versa.**” (Lines 136-155)*

To answer the reviewer; the model does not contain explicit multisynaptic connections. Instead, the maximal conductance of each synapse is a bit different, simulating more or less synaptic sites onto the same single-compartment neuron.

R3.9. L154 consequently, an experiment with NMDAR blockers and measuring short-term synaptic depression on this system might be proposed in the discussion for electrophysiologists to test the predictive capacity of the approach

Our results (with the sensitivity test of NBD) and previous pharmacological tests (Doorn et al., 2024b; Frega et al., 2019) suggest that both NMDAR conductance and the strength of short-term synaptic depression can inversely modulate the NBD feature. These parameters may therefore exhibit a high conditional correlation in the model.

We note that such correlation does not necessarily imply a biological compensation mechanism between NMDAR function and short-term depression. Even if no such link exists, this does not limit the predictive capacity of our approach as inference is based on matching observations rather than assuming compensatory mechanisms. An experiment combining NMDAR blockage with a readout on short-term synaptic depression could indeed help confirm the existence of a compensatory mechanism.

SBI can identify pharmacological targets ...

R4.1. L171 synaptic vesicle recycling (COMMA) thereby

Added

R4.2. L176-L177 consider metrics such as e.g. OVL (see above) for quantification at the level of the posterior

We thank the reviewer for the suggestions. We have now considered OVL. We found, however, that there currently exists no way to specify a cut-off value for the OLV (i.e., when is a distribution so different that we should look into it experimentally). With the KS-statistic, it feels natural to take the 0.05 p-value value as a cut-off. Thus, we opt to stick to the KL-statistic. Nevertheless, we agree that OVL, to at least compare parameter changes relative to each other, would be a reasonable alternative.

R4.3. L178 simulations with the mode of EACH [not BOTH] posterior distribution (...) appear highly similar to the CORRESPONDING

Changed. (Line 180)

R4.4. L184 I am not a statistician, but it is unlikely that averages of a nonlinear measure, such as a *p*-value are very meaningful, and they are most certainly not a p-value that can be compared (dashed line) to typical thresholds of significance such as 0.05. I can think of two solutions:

1. explore multiple K-S tests (see <https://stats.stackexchange.com/questions/35461/is-there-a-multiple-sample-version-or-alternative-to-the-kolmogorov-smirnov-test> for references)
 2. or, instead of spending effort in sophisticated averaging of *p*-values (e.g. as outlined in <https://www.semanticscholar.org/paper/Combining-P-Values-Via-Averaging-Vovk-Wang/29c9b82c9fd88d2e9dba99ad79301a281916aee6>), I suggest if the number of *p*-values is reasonable, to use a simple per-parameter scatter plot for Fig 3D and similar (i.e. use the jittering approach described above). The distributional properties of the *p*-values are more faithfully represented than with typical bar and error bar plots (in particular, extreme values can be spotted), and they can all be individually compared with significance thresholds just fine. The interpretation in the text does not need averages either.
 3. In any case, for readability, you may want to plot the range [0.0 - 0.05] in an inset with enlarged scale, or use a logarithmic scale for the y-axis, in which case you can add dashed lines at *, **, ***, etc significance levels to guide the eye, relevant to Fig 4
- 
To check whether averaging p-values here was inherently wrong, we consulted with a mathematician. They found the averaging reasonable in this case. But we agree with the reviewer that average p-values might not be as meaningful as the distribution of the individual p-values. We changed the figure to show the individual data points and adapted the text accordingly. Unfortunately, the use of an inset or logarithmic scale made the figure quite unreadable, so we decided against this.

In the text, we have removed the notion of average p-values:

“To investigate whether these parameter changes were consistent among many networks, we evaluated the significance of these changes for all networks (Fig. 3D).”
(Line 187)

Figure 8: New panel D of Figure 3 (same change for Figures 3H, 4C, and 4F). Now we show scattered p-values and a line indicating $p=0.05$, and we removed the star to denote that the average $p\text{-value} < 0.05$. Changed caption: *“Average p-values of the KS-test result of 2 batches per parameter. ... Dashed line indicates $p=0.05$ ”*

4. **L198 the conclusion is not complete: perhaps the summary features do not change significantly or consistently but the underlying neuronal network activity pattern does. Besides acknowledging this possibility (which authors do perfectly in L262 in the discussion), there are two ways to improve here: one is to quantify the changes induced by MK-801 at the level of features; the other is to (as suggested above) use more comprehensive activity fingerprinting, either via PCA and retaining many components or even better, as suggested above and explained below, by adopting an embedding network**

To quantify the changes induced by MK-801 at the level of the features, we used the Energy distance:

Comparison	Posterior Figure	Energy Distance
Healthy 1 vs. Healthy 2	Figure 2B vs. Supp Figure 2B	5.25
Before vs After Dynasore	Figure 3A	3.64
Before vs After MK801	Figure 3E	0.71
Control vs DS	Figure 4A	6.04
Control vs GEFS+	Figure 4A	2.44
Control vs CACNA1A ^{+/-}	Figure 4D	1.31

The distance between MEA features of the observation before and after applying MK801 is almost a factor 5 smaller (ED=0.71) than that of Dynasore (ED=3.64). This is consistent with the normalized MEA features shown in (Figure 3F), where most values

remain close to 1, indicating only minor changes after MK-801. Additionally, there is a large variability across networks: in some, MK801 increases NBR; in others, it decreases it. This is what we meant when stating that the features do not change consistently, unlike with Dynasore, which uniformly shortens the NBD across all networks.

We fully agree with the reviewer that it is possible that the consistent changes in network activity are not captured by the 13 hand-crafted features. In our current implementation, the embedding network yielded posterior distributions that were too broad to allow meaningful comparisons. Despite this limitation, we have made this more explicit in the revision:

*“Using MK-801, an NMDAR-blocker, SBI predicted a decrease in connectivity, but this prediction was not consistent across all networks. Moreover, SBI predicted an initial NMDAR conductance close to zero. This appears in line with the lack of a clear effect of MK-801 on the MEA activity features in most networks. **This shows that SBI, as well as other parameter inference techniques, can only identify changes that are already clearly and consistently reflected in the characterization of system behaviour used as input. Our MEA features might not fully capture the change in network activity induced by MK-801.**” (Lines 263-266)*

SBI pinpoints known disease mechanisms ...

R5.1. L205 authors say “because we previously observed that different batches affect SBI results...” it is not clear how much observations from different batches already differ at the level of the 13 input features.

See response to R2.14.

R5.1. Fig S2B and Fig 3: the variability across batches is transparently presented in order to contextualize the putative disease-driven or genetic-manipulation-induced variability, but what about the inter-individual variability? Could a second control subject could help understand this driver of variability? The drug experiments are both based on cell lines from the same subject, so this comment does not apply there

Our previous research into these types of networks showed that many factors, like astrocyte batch and seeding density, caused variability in activity (Mossink et al., 2021). Different healthy subjects, however, showed relatively similar neuronal network activity. Nevertheless, there was some variability between healthy subjects, and it could, therefore, be wise to include more control subjects when studying patient lines. Unfortunately, in our dataset, there were no multiple controls cultured on the same plate, as discussed before.

R5.2. Fig. 4C and Fig. 4F --- same comments as above apply to aggregation of p^* -values, but here it is really critical to reconsider the approach because the stars may give the impression that the average p^* -value is a p^* -value, which is not

We have removed all notions of averaged p-value as explained in the response to R4.4

R5.3. L231 same comment as above where it is not clear that number of synapses stands for Conn%

Here, it concerns *in vitro* experiments where there was a lower number of synapses found by immunostaining. So in L231 (now L233), we are not referring to the number of synapses in the model.

Discussion

This section compares the changes observed in SBI-estimated parameters after a number of interventions both to known mechanisms and direct, independent experimental findings in vitro, and makes an overall convincing case for the authors' use of SBI and for the field to adopt it

D1. L239 the phrase can be improved to not use a generic word like utilizing or using (“MEAs allow quick collection of ephys data from hiPSC-NNs”). If you insist on utilizing, however, I find these stylistic remarks appropriate: <https://www.merriam-webster.com/grammar/is-utilize-a-word-worth-using>

Changed to:

“MEAs provide a rapid means to gather neuronal electrophysiological data from patient-derived or genome-edited neuronal networks.” (Lines 241-242)

D2. L245 for the expressions: RNA sequencing, time consuming: don't join words with a dash; they would receive one if they preceded a noun, such as experiments

We were unaware of these grammar rules and thank the reviewer for pointing them out, thereby improving our overall writing. We altered this.

D3. L253 WHEN using Dynasore (...) [otherwise it seems that SBI was using Dynasore]
Adapted

D4. L262

D4.1. “clearly” is obvious (no analysis procedure can speculate phenomena that leaves no trace in their input data) so that it might invite the impression that there is a limitation of SBI at play. But this is not the case - SBI is flexible to accommodate more comprehensive and more data-driven summaries of observations as inputs -- as described above. I suggest to rewrite along the lines “parameter inference (and hence SBI) can only identify changes that are already clearly and consistently reflected in the descriptors of system behavior that constitute its input”

Changed to:

“This shows that SBI, as well as other parameter inference techniques, can only identify changes that are already clearly and consistently reflected in the characterization of system behaviour used as input” (Lines 263-265)

D4.2. “consistently” begs the question of the relative impact of batch-specific factors (nuisance factors) wrt. all others on the network activity. It would aid understanding to quantify how interventions impact MEA features vs. just drawing from another batch under no intervention (same for drawing from another control individual, same batch).

Co-author dr. Frega investigated and quantified this earlier wrt to most MEA features in Mossink et al. (Mossink et al., 2021). This reference is included in the manuscript.

D5. L273 - since forecast has connotations relating to time I suggest to use estimate instead (there is no before/after the disease with a comparison in the same individual); further, there is no “increase” because there is no temporal ordering ---it would be more appropriate to write that SBI estimates a higher STD under this or that intervention ... Accordingly, predictions -> inferences in L275

Changed:

“Additionally, SBI estimated significantly higher values of STD strength in both GEFS+ and DS, along with a larger STD time constant for DS and increased asynchronous release in GEFS+. These inferences align...” (Lines 276-278)

Limitations and advantages

L1. L317 - here misspecification is discussed in a potentially confusing way. As highlighted in Major comment 1, it must be taken for granted that the model is very strongly misspecified with respect to the in-vivo network. Yet, this does not detract from its main conclusions 1) that SBI is a great advancement in parameter inference in biology, and 2) that results leveraging the authors' simulator are largely in agreement with known mechanisms and direct, in-vitro measurements on comparable model systems. In line with the initial comment, I suggest to clearly state that one should be cautious in drawing conclusions from an entirely in-vitro framework for the living system, insofar as the hiPSCs-NNs are a relatively simplified model for it, and that in any computational investigation of hiPSCs-NNs the computational model (the simulator, here) may not fully represent the dynamics even if they are already much simplified in comparison to in vivo. So the misspecification that is not trivial and matters to highlight is that of the simulator and the in-vitro model, not the obvious and likely insurmountable one if the simulator was attempting to model the in-vivo behaviour, which is not.

Yet, the results speak for themselves and support the publication of this paper, with all the provided caveats and potentially with more work to quantify misspecification, leading to an improved model in the future and warrant a nuanced cautious recommendation to widely adopt SBI in the community.

L2. L324 I am unconvinced that a more involved interpretation is a price to pay for potentially fitting a misspecified, because overconstrained (via fixed parameters), model, and I offered a suggestion elsewhere. Regarding the computational burden of the simulator, I do agree it represents a practical problem as parameter numbers increase, but for the sake of future users of their simulator, I would like the authors to give representative orders of magnitude of simulation times in the Methods.

Before settling on this final model, we tried models with different fixed and non-fixed parameters. There, we noticed, for example, that if both synaptic time constants and conductances were “free” parameters, we would almost never see obvious changes in their univariate marginals because the two can compensate for each other to a large extent. The same goes for, for example, the time constant and conductance of the AHP current. Marginals got wider, even on synthetic data, and differences between conditions were much less obvious. We agree that it would be more elegant to train with all parameters as free parameters and condition afterwards, but we could not do this computationally.

We now mention the simulation times on our system in the methods:

“One simulation took about two to three times the simulated time on a regular desktop CPU.” (Lines 487-488)

L3. L328 this might be a plausible argument in terms of pathophysiological detection or classification, but then we don't need a mechanistic simulator and sophisticated inference engine: just the embedding network discussed elsewhere coupled to a classifier would do. The danger of a misspecified model is that some parameter not causally linked to the particularities of an observation will be tuned to best match it, because the really responsible parameter is not free to vary.

Indeed, the goal of the model is not to detect differences between networks but to find out the pathophysiological mechanisms underlying them. We changed this:

“Additionally, it could complicate the interpretability of the inferred posterior distributions, due to parameter redundancy and the widening of marginals” (Lines 333-335)

L4. L337 I appreciate the introduction of additional features in search of a sufficiently sensitive set

L5. L341 sounds like a concrete, actionable, feature request for the simulator or the MEA summaries. If the current simulator cannot match the in-vitro typical signal-to-noise ratios (SNR) in silico, it should be structurally improved; alternatively, if it is sufficiently expressive already, then, the SNR might constitute a suitable 14th handcrafted MEA summary feature?

The signal-to-noise ratio differences are mainly caused by bad MEA electrodes and are thus not informative about the physiological parameters of the neuronal network. In general, our model does simulate a proper SNR if the measurement electrode is perfect, because the model includes membrane potential noise. However, we do not want to fit additional measurement noise because it is not informative about the biophysical parameters.

L6. L344-348 note that the sbi library bundles more and more recent posterior checks, such as TARP (complete list here: <https://sbi-dev.github.io/sbi/latest/#diagnostics>)

We added these references:

“Recently, more posterior checks have become available that could be employed by future users to further assess the validity of inference^{37,38}” (Lines 371-372)

L7. L353-368 this paragraph articulates very clearly why SBI is the most compelling approach for parameter inference on computational models of biological function, and why this paper deserves publication at this venue. Consequently, I find the last sentence (L367-368) makes a too conservative summary of the paragraph and the author's comprehensive work. Based on this and the rapidly growing body of applied literature across the most diverse fields of science, SBI (not just posterior analysis, the posterior may be inferred in other ways) not just "could" be a "plausible addition" to the MEA analysis pipeline, but it constitutes the most appealing of the (already more compelling) Bayesian approaches, sometimes the only plausible in high-dimensionality parameter spaces.

Changed to:

“Therefore, SBI represents the most compelling approach for integrating parameter inference into the standard MEA data analysis pipeline.” (Line 394)

Recommendations

Rec1. L374 note that if those confounding factors could be mechanistically modeled within the simulator, it would be possible either to co-infer e.g. the astrocyte count or where experimentally feasible, to measure them for each batch and have batch-specific simulators to account for batch-local conditions.

Agreed.

Rec2. L379 a SUFFICIENT NUMBER/QUANTITY of batches

Rec3. L381 I believe (see above) that averaging p-values is unsound statistical advice and that it suffices, in absence of a more principled procedure, the authors should limit their recommendation to drawing conclusions based on the analysis of a sufficiently representative amount of batches

Rec4. L383 SBI-identified (see above)

Rec5. L384 this is meanwhile not anymore correct, see above

For Red2-Rec5, we changed:

“Previous advice emphasized the importance of employing a sufficient number of experimental batches², and here we observed that not all parameter differences were reproduced across batches. We thus strongly recommend performing sufficient experimental replicates and comparing univariate marginals per MEA to identify parameters that are altered consistently. Thirdly, targeted in vitro investigations of SBI-identified disease mechanisms are imperative to validate predictions. In some cases, invalid inferences could result from undetected model misspecification. Additionally, SBI...” (Lins 405-411)

Methods

Computational model

M1.1. The numerical integration of the model is not discussed. If using forward / exponential Euler like in previous publications (e.g. Doorn et al. 2023, Stem Cell Rep.), how has it been ensured that there is no accumulation of error over the integration time? Is there an option in Brian2 to employ a higher-order method, e.g. from the (stochastic) Runge-Kutta family with better stability properties?

Brian2 does include the classical Runge-Kutta method, but this does not support stochastic differential equations, while we could easily include the approximate noise term in our model solved with Exponential Euler. Other methods, like Heun and Milstein were unstable for our model. We ensured the stability of our integration by comparing our simulations to simulations with much smaller timesteps and by comparing simulations (without noise) to those solved with the RK method. Moreover, the Exponential Euler produced consistent and biologically plausible results without divergence over long simulations.

M1.2. The citation to the synaptic-parallel computing engine is obscure: it is unclear whether the use of this (custom?) platform is merely for speeding simulations up, or if reproducibility of the work with standard computers would be hampered. It is difficult to understand the platform by reading the appendix of an astrophysics paper (reference 40). Could the authors briefly indicate what prompts the decision to use the platform, and if any specifics of it are liable to introduce differences with a standard serial architecture? The published code, upon cursory inspection, does not show traces of the use of any non-standard hardware

We understand this might be confusing. Because we do not have access to a high-performance computer cluster within our faculty, we accelerated our forward simulations with a custom-made parallel computing cluster that combines several different desktop machines. This dedicated platform features dynamic load balancing by a synaptic parallel computing under bash, recently developed for observational astronomy. Different from conventional master-slave configurations, the worker nodes initiate requests for tasks from a shared process list with atomic access. This allowed us to add any lab desktops to the cluster as they become available, even during cluster run-time, maximizing the use of our resources. We replaced the reference to the astrophysics paper with a reference to the patent application (van Putten, 2023), which more clearly explains the method:

“To parallelize simulations across multiple desktop computers, we used a distributed computing method with dynamical load balancing⁴⁸. While this platform significantly accelerates computation and facilitates flexible resource usage, it is not essential to the simulations themselves—similar performance could, in principle, be achieved with alternative parallel computing architectures.” (Lines 488-492).

Preprocessing

M2.1. While the Butterworth filter, being an IIR filter, is fast and can be used online, FIR-type filters such as Parks-McLellan are preferable in situations where the whole signal is available at once, such as is the case here (but FIR filters have border effects, so that the beginning and end of the recording are not directly usable). Another reason to promote FIR filters is how easily they compose with smoothing kernels, see next point.

M2.2. For more elegant signal detection, consider merging the low-pass filter used initially with subsequent phases of time-domain aggregation/averaging

1. **computation of firing rates over bins is equivalent to convolution with a square-window convolution kernel (which is spectral properties)**
2. **smoothing of firing rates with a Gaussian for fragment detection is yet again another low-pass filter with a Gaussian kernel**
3. **... so you can just design composite filter kernels and use the once for the respective features**

The low-pass filter used initially was applied to the raw voltage traces, while the computation of firing rates happened after spike detection. Therefore, we believe these “filters” cannot be merged. But indeed, binning spikes and smoothing the firing rate could be combined, and we will take this into account for future work. We thank the reviewer for the suggestion.

4. **ideally the peak detection algorithm parameters should be discovered by systematic comparison with human-labeled spike trains (e.g. by training a segmenter or classifier model). But at that point it might be better to use the end-to-end approach discussed below**

Peak detection was compared using an informal comparison to human labels, which, admittedly, is not very rigorous. In general, the algorithm used for detecting network bursts (NB)—a commonly used feature in the MEA community—relies on somewhat arbitrary parameters, such as detection thresholds and prominences, which lack a principled definition.

The other features in our set do not depend on such heuristics. They are generally parameter free or based on well-defined rules, making them more robust and less sensitive to subjective choices. While one could argue for excluding NB features due to their less well-defined parametrization, we chose to retain them because they are widely recognized and routinely used by MEA users, and have been shown to be associated to (patho)physiological mechanisms.

5. **the composite-filter approach can also be done on 2D to account for spatial features of the MEA firing pattern**

M2.3 But, overall, the main weakness is the selection of summary features and the elaborate hand-calibrated detection, with obscure settings such as 1/16th peak height and 1/20th prominence, etc, which emerge from tedious visual inspection by an expert. What I would like to see to strengthen results, is an end-to-end approach, where the entire signal is processed by a convolutional network (1D) over the spike-time series, potentially with skip connections, but has electrode exchange invariance (i.e. it handles electrodes as a set). As reasoned above, a number of summary features based on more or less instantaneous rates emerge naturally from the hierarchical-convolutional approach. The approach can be applied for the ISI time-series just as well (or to a frequency-domain representation of short-term rates), and the outputs of these summarizing networks can be concatenated for further downstream processing

1. **if the spatial structure of firing times and intervals (e.g. for SD ISI elec) might be important, this should be done over 2D and there is no need to introduce exchange**

invariance among electrodes; if the only scale relevant to the phenomenon is of the size of the MEA array (“network” events), this will be picked up by the convolutional network by using appropriately wide convolution kernels

2. researchers in [6] used an autoencoder to find summary statistics to then feed to SNPE. They pretrained it by forcing it to reproduce synthetic data and then refined the encoder part together with the training of the NDE (a MAF). Sections II-C and II-D contain the details (note I interpret they want to say 'jointly' instead of 'in parallel'). This could be a useful template to follow in order to get data-driven summary features. The library `sbi` already has all the facilities in place to use an embedding network
3. this represents some additional effort, and not doing it, in my view, is not a showstopper for publication, but a more comprehensive and critical engagement with the selection and extraction of summary features would provide a qualitative leap in the quality of the paper and trustability of results, so I would also value any efforts to render the current selection more convincing in the absence of a more structured approach -- authors discuss critically the limitations of hand-crafting summary features indeed, but they could also explore the sensitivity of results against the inclusion of redundant features or the removal of (presumed) key ones
4. L484 - the authors initially trained the NDE with additional features that they then argue can be discarded because based on a high value (0.96) of correlation (a linear measure of association), they see those as redundant. However, I believe that the stronger argument, given that they already have trained the NDE, would be based on the distributions (or at least their modes) of the SBI-derived parameters. Adding redundant inputs should not affect those significantly. To what extent is this the case?

Please see the response to major comment 3.2.2 about our rationale for choosing these MEA features.

Additionally, when starting this project, we informally assessed the influence of adding or removing certain MEA features from training. We did so by training several NDEs and comparing posterior distributions on the same data.

Network features were our initial focus as they are the most widely used in studies involving hiPSC-derived neuronal networks on MEA, and are well-known across the field. As discussed before, several associations between NB features and molecular (patho)mechanisms have been shown.

We noticed, however, that in the absence of NBs in the observations, these features were not very informative, and inferred marginals were wide. To address this, we searched literature for alternative MEA features that do not rely on network bursting. Using visual analysis and synthetic data, we observed that including these non-NB features in the training of the NDE improved inference.

We like to emphasize that we never removed the additional two features (with high correlation to other features) from training of the NDE used for the results in our manuscript; we simply did not report these features explicitly as they were so similar to the other ones.

Now, we performed a slightly more systematic assessment of the effect of adding or removing features on inference performance. We drew 25 new samples from the prior distribution to generate synthetic data, performed inference on that data with NDEs trained on four different subsets of features (all features, only NB features, only non-NB features and without CC features), and computed the PRE for each. Below, example posterior distributions can be seen for one synthetic input, along with violin plots for all PRE values.

Figure 9: A-D Posterior distribution inferred from synthetic data with the NDE trained on A) All 15 MEA features; B) Only the 5 NB feature; C) All features except the 5 NB features; and D) All 13 features without the Correlation Coefficient features. E) Simulation used as input (ground truth), and simulations with the modes of the posterior distributions shown in A-D. F) Parameter Recover Error (PRE) values of 25 evaluations with synthetic data. G) Average PREs over all simulations. 2-Way ANOVA with Tukey's multiple comparisons test was performed. ns $p > 0.05$, * $p < 0.05$, ** $p < 0.01$.

It can be seen from this figure that, as we expected, the network trained on only the NB features exhibits higher PRE values throughout compared to the network trained on all 15 features. It is important to note, however, that the NB feature set contains only 5 features, whereas the non-NB feature set includes 10. Somewhat unexpectedly, the network trained on these remaining 10 non-NB features performed similarly to the one trained on all features. Thus, from this analysis, it seems the addition of the NB features does not help inference and could therefore have been omitted from training. Nonetheless, these 5 NB features are the most well known and widely adopted in the field. As such, they remain valuable for reporting purposes, especially when comparing health and disease states, or *in silico* and *in vitro* results, to ensure consistency with the literature and facilitate interpretation by domain experts.

We included the comparison shown in Figure 9 in a new Supplementary Figure 5, which we refer to in the discussion:

“Additionally, we assessed the performance of SBI with NDEs trained on different subsets of MEA features (Supplementary Fig. 5). This showed that while NB-detection dependent features do not harm inference, the addition of the other features improves inference.” (Lines 364-367).

Removing features 14 and 15 (that had high correlations to other features) did not significantly impact PRE values (see trained NDE without CC features in Figure 9).

Conditional correlations

M3. To detect the presence of complex relationships in conditional marginal distributions, authors would need to go beyond linear associations as captured by correlation coefficients and compute measures of mutual information (see third row of first figure in <https://en.wikipedia.org/wiki/Correlation>). This can only strengthen their argument in general, even if in most of the conditional marginals shown the association is relatively well-captured by a linear relationship, i.e. based on data shown, I do trust the overall thrust of the argument

Indeed, some associations between parameters might not be linear, and calculating something like mutual information (MI) might even show stronger associations than those found with correlation coefficients in these cases. Therefore, we attempted to calculate it. We tried to alter the sbi toolkit functions that automatically calculate conditional correlations (https://github.com/sbi-dev/sbi/blob/main/sbi/analysis/conditional_density.py) to instead compute conditional mutual information.

However, we noticed MI estimation is sensitive to numerical artifacts. Despite our effort to try multiple methods to estimate MI (KNN-based estimator, Kraskov-Stögbauer-Grassberger estimation, Nonparametric Estimator), we kept encountering numerical difficulties in many conditionals, and MIs could never be estimated for all parameters in all conditions.

Our goal of showing conditional correlations in this manuscript was to show that there are degeneracies and that, therefore, SBI is more useful than methods that estimate only one set of optimal parameters. We also wanted to show that these degeneracies align with our intuition and knowledge about the biological system. Our goal is not to draw conclusions about specific associations that exist between parameters or the magnitude of these associations. Given the stable and interpretable results from correlation coefficients and the numerical difficulties of MI estimation in our setting, we use correlation as a more robust and reliable measure.

Good work! I enjoyed reading the paper.

We again would like to thank the reviewer for all the time and effort put into this review. It helped us a lot, and we hope our responses are satisfactory.

Kind regards, Álvaro Tejero-Cantero.

References by Reviewer #3

- [1] [[2412.15100] Tests for model misspecification in simulation-based inference: from local distortions to global model checks](<https://arxiv.org/abs/2412.15100>)
- [2] [[2406.03154] Detecting Model Misspecification in Amortized Bayesian Inference with Neural Networks: An Extended Investigation](<https://arxiv.org/abs/2406.03154>)
- [3] [[2209.01845] Investigating the Impact of Model Misspecification in Neural Simulation-based Inference](<https://arxiv.org/abs/2209.01845>)
- [4] [Methods and considerations for estimating parameters in biophysically detailed neural models with simulation based inference | bioRxiv](<https://www.biorxiv.org/content/10.1101/2023.04.17.537118v1.full>)
- [5] [[2412.02437] Reproduction of AdEx dynamics on neuromorphic hardware through data embedding and simulation-based inference](<https://arxiv.org/abs/2412.02437>)
- [6] [Real-Time Gravitational Wave Science with Neural Posterior Estimation | Phys. Rev. Lett.](<https://journals.aps.org/prl/abstract/10.1103/PhysRevLett.127.241103>)
- [7] [Uncertainty mapping and probabilistic tractography using Simulation-Based Inference in diffusion MRI: A comparison with classical Bayes | bioRxiv](<https://www.biorxiv.org/content/10.1101/2024.11.19.624267v1.full>)
- [8] [Simulation-based inference on virtual brain models of disorders | IOPscience](<https://iopscience.iop.org/article/10.1088/2632-2153/ad6230/meta>)
- [9] [Pathological cell assembly dynamics in a striatal MSN network model | Frontiers](<https://www.frontiersin.org/journals/computational-neuroscience/articles/10.3389/fncom.2024.1410335/full>)
- [10] [Amortized Bayesian inference on generative dynamical network models of epilepsy using deep neural density estimators | ScienceDirect](<https://www.sciencedirect.com/science/article/pii/S0893608023001752>)
- [11] [Combined statistical-mechanistic modeling links ion channel genes to physiology of cortical neuron types | bioRxiv](<https://www.biorxiv.org/content/10.1101/2023.03.02.530774v1.full#sec-9>)
- [12] [Bayesian Inference of a Spectral Graph Model for Brain Oscillations | bioRxiv](<https://www.biorxiv.org/content/10.1101/2023.03.01.530704v2.full#F5>)
- [13] [The virtual aging brain: a model-driven explanation for cognitive decline in older subjects | bioRxiv](<https://www.biorxiv.org/content/10.1101/2022.02.17.480902v2.full>)

[14] [Fast inference of spinal neuromodulation for motor control using amortized neural networks | IOPscience](<https://iopscience.iop.org/article/10.1088/1741-2552/ac9646/meta>)

[15] [Inferring Morphology of a Neuron from In Vivo LFP Data | IEEE Conference Publication | IEEE Xplore](<https://ieeexplore.ieee.org/abstract/document/9441161>)

[16] [Training deep neural density estimators to identify mechanistic models of neural dynamics | eLife](<https://elifesciences.org/articles/56261>)

[17] [Fast ϵ -free Inference of Simulation Models with Bayesian Conditional Density Estimation | Neurips](<https://proceedings.neurips.cc/paper/2016/hash/6aca97005c68f1206823815f66102863-Abstract.html>)

[18] [Benchmarking Simulation-Based Inference | AISTats](<https://proceedings.mlr.press/v130/lueckmann21a.html>),
([Appendices](<https://proceedings.mlr.press/v130/lueckmann21a/lueckmann21a-supp.pdf>))

Rebuttal References

- Ardizzone, L., Lüth, C., Kruse, J., Rother, C., Köthe, U., Learning, V., & Heidelberg, L. (2019). *Guided Image Generation with Conditional Invertible Neural Networks*. <https://arxiv.org/abs/1907.02392v3>
- Doorn, N., van Hugte, E. J. H., Ciptasari, U., Mordelt, A., Meijer, H. G. E., Schubert, D., Frega, M., Nadif Kasri, N., & van Putten, M. J. A. M. (2023). An in silico and in vitro human neuronal network model reveals cellular mechanisms beyond NaV1.1 underlying Dravet syndrome. *Stem Cell Reports*, *18*(8), 1686–1700. <https://doi.org/10.1016/J.STEMCR.2023.06.003>
- Doorn, N., Voogd, E. J. H. F., Levers, M. R., van Putten, M. J. A. M., & Frega, M. (2024a). Breaking the burst: Unveiling mechanisms behind fragmented network bursts in patient-derived neurons. *Stem Cell Reports*. <https://doi.org/10.1016/J.STEMCR.2024.09.001>
- Doorn, N., Voogd, E. J. H. F., Levers, M. R., van Putten, M. J. A. M., & Frega, M. (2024b). Breaking the burst: Unveiling mechanisms behind fragmented network bursts in patient-derived neurons. *Stem Cell Reports*. <https://doi.org/10.1016/J.STEMCR.2024.09.001>
- Frega, M., Linda, K., Keller, J. M., Gümüç-Akay, G., Mossink, B., van Rhijn, J. R., Negwer, M., Klein Gunnewiek, T., Foreman, K., Kompier, N., Schoenmaker, C., van den Akker, W., van der Werf, I., Oudakker, A., Zhou, H., Kleefstra, T., Schubert, D., van Bokhoven, H., & Nadif Kasri, N. (2019). Neuronal network dysfunction in a model for Kleefstra syndrome mediated by enhanced NMDAR signaling. *Nature Communications*, *10*(1), 1–15. <https://doi.org/10.1038/s41467-019-12947-3>
- Frega, M., Van Gestel, S. H. C., Linda, K., Van Der Raadt, J., Keller, J., Van Rhijn, J. R., Schubert, D., Albers, C. A., & Kasri, N. N. (2017). Rapid neuronal differentiation of induced pluripotent stem cells for measuring network activity on micro-electrode arrays. *Journal of Visualized Experiments*, *2017*(119), 54900. <https://doi.org/10.3791/54900>
- Hommersom, M. P., Doorn, N., Puvogel, S., Lewerissa, E. I., Mordelt, A., Ciptasari, U., Kampshoff, F., Dillen, L., van Beusekom, E., Oudakker, A., Kogo, N., Dolga, A. M., Frega, M., Schubert, D., van de Warrenburg, B. P. C., Nadif Kasri, N., & van Bokhoven, H. (2024). CACNA1A haploinsufficiency leads to reduced synaptic function and increased intrinsic excitability. *Brain*, *139*(4), 16–17. <https://doi.org/10.1093/BRAIN/AWAE330>
- Lenk, K., Satuvuori, E., Lallouette, J., Ladrón-de-Guevara, A., Berry, H., & Hyttinen, J. A. K. (2020). A Computational Model of Interactions Between Neuronal and Astrocytic Networks: The Role of Astrocytes in the Stability of the Neuronal Firing Rate. *Frontiers in Computational Neuroscience*, *13*, 480462. <https://doi.org/10.3389/FNCOM.2019.00092/BIBTEX>
- Maheswaranathan, N., Ferrari, S., VanDongen, A. M. J., & Henriquez, C. S. (2012). Emergent bursting and synchrony in computer simulations of neuronal cultures. *Frontiers in Computational Neuroscience*, *6*(MARCH 2012), 1–9. <https://doi.org/10.3389/FNCOM.2012.00015/BIBTEX>
- Mossink, B., Verboven, A. H. A., van Hugte, E. J. H., Klein Gunnewiek, T. M., Parodi, G., Linda, K., Schoenmaker, C., Kleefstra, T., Kozicz, T., van Bokhoven, H., Schubert, D., Nadif Kasri, N., & Frega, M. (2021). Human neuronal networks on micro-electrode arrays are a highly robust tool to study disease-specific genotype-phenotype correlations in vitro. *Stem Cell Reports*, *16*(9), 2182–2196. <https://doi.org/10.1016/J.STEMCR.2021.07.001>
- Schmitt, M., Bürkner, P.-C., Köthe, U., & Radev, S. T. (2024). *Detecting Model Misspecification in Amortized Bayesian Inference with Neural Networks: An Extended Investigation*. <https://arxiv.org/abs/2406.03154v2>
- Van Hugte, E. J. H., Lewerissa, E. I., Wu, K. M., Scheefhals, N., Parodi, G., Van Voorst, T. W., Puvogel, S., Kogo, N., Keller, J. M., Frega, M., Schubert, D., Schelhaas, H. J., Verhoeven, J., Majolie, M., Van 5 Bokhoven, H., & Kasri, N. N. (2023). SCN1A-deficient excitatory neuronal networks display mutation-specific phenotypes. *Brain*, *146*(12), 5153–5167. <https://doi.org/10.1093/BRAIN/AWAD245>
- van Putten, M. H. P. (2023). *Method for high-throughput load balanced data-processing in distributed heterogeneous computing* (Patent US 20240303120A1). van Putten, Maurice Hendrikus Paulus.

Nina Doorn
Drienerlolaan 5
7522 NB Enschede
The Netherlands

Telephone: +31611398957
n.doorn-1@utwente.nl

Enschede, 02-05-2024

We want to sincerely thank the reviewers for their encouraging and kind words and for deeming our manuscript significantly improved and suitable for publication. Below, we address the remaining suggestion of Reviewer 3.

Reviewer #3 (Remarks to the Author):

First I want to thank you, the authors, for an excellent rebuttal text, that so much facilitated my review.

The revised manuscript is, to put it shortly, impressive. The authors introduce multiple new analyses in support of their original contribution, some of them representing a very substantial effort and all of them relevant to substantiating the discussion we have been having in a satisfactory manner.

A particularly salient example is the training of a dedicated embedding network. This analysis offered a valuable negative result, which is worthy of sharing with the SBI community and discussing in the appropriate venues.

While I believe the authors correctly appreciate that including Rebuttal Fig. 4 in the article might sidetrack the neuroscience-minded reader of the present journal, I would suggest to them, and the Editor, to consider adding that Figure and relevant paragraphs in Appendix or, at their discretion, briefly mention in the text how their experiments with embedding networks (necessarily limited!) resulted in inconclusive, washed-out posteriors at an elevated inference cost.

The interest of reporting this effort (whether in the present publication, or elsewhere) lies in how it might spark a discussion in the SBI community about practical approaches to combine the strength of expert-vetted features and the flexibility of neural network-based feature identification, which is a theme I have thought about (SBI takes the path not of replacing mechanistic modeling with expressive, but blind, neural networks, but of using the latter in the service of our hard-won understanding of systems as condensed into mathematical laws and calculated using simulators. Similarly, we should strive for neural networks not to replace summary statistics, something that, as Fig. 4 intimates to us, can be a hard and costly program, but to improve them.

The MMD assessment, newly computed PRE values, “ablation” analysis of the impact on posteriors of burst-related features, energy distance analysis, and the clarifications of intent and language, all further contribute to the solid and rigorous piece of work.

In sum, the paper in current form represents a strong contribution to the field of MEA analysis and an exemplary practical application of SBI techniques to the advancement of science, with a clear and pedagogical message about interpretational pitfalls and limitations. I sincerely hope that it will be widely read by both communities and that future applied SBI work takes inspiration from it.

Response:

To allow the community insight into the negative result obtained with the embedding network, we have now included the figure in the manuscript supplementary information as supplementary figure 5, with a caption shortly describing the methodology and main takeaways:

“Figure S5: **Comparison of posterior inference using hand-crafted versus data-driven summary features.** **A)** Posterior distribution inferred from synthetic data using a neural density estimator (NDE) trained on hand-crafted MEA features. **B)** Posterior distribution inferred from the same synthetic data using an NDE trained jointly with a 2D-convolutional embedding network, applied to multi-unit firing rate data. The resulting marginals are noticeably wider and less informative than in A. **C)** Raster plots showing 1 minute of simulations with the ground-truth parameters and the modes of the posterior distributions depicted in panel A and B. While the simulation resulting from the MEA-features approach resembles the bursting behavior of the synthetic data, the simulation with the embedding-network approach shows little similarity to either. This result illustrates a negative finding: despite efforts to develop and train a suitable custom embedding network, parameter recovery was less accurate than with expert-defined features. The average parameter recovery error with hand-crafted MEA features was 0.0728, compared to 0.1670 with the embedding network.”

Additionally, we have included a supplementary methods section to outline the approach taken with the embedding network:

“To assess whether data-driven summary statistics could improve parameter inference, we implemented a convolutional embedding network trained jointly with the neural density estimator (NDE). Our goal was to explore whether such embeddings could outperform our hand-crafted MEA features.

We generated 100,000 simulations with parameter configurations sampled from our box prior. Input to the embedding network consisted of multi-unit firing rates binned at 100 ms over a 3-minute recording, resulting in input tensors of size 12 (electrodes) \times 1800 (time bins).

The embedding network was designed to extract both local spatio-temporal features and long-range temporal dependencies. Its architecture included:

1. A 2D convolutional layer (kernel size 3×5) to capture local electrode-time patterns;
2. A depthwise convolution (1×5 kernel) to further model temporal structure;

3. A 1×1 convolution for integrating information across electrodes;
4. Two dilated convolutions (1×5 kernel, dilation rates 2 and 4) to capture long-range dependencies;
5. Two max-pooling layers (1×2 kernel) for dimensionality reduction;
6. A flattening layer, followed by a fully connected ReLU layer (256 units), and a final linear projection to a 20-dimensional embedding space.

This embedding network was trained jointly with the NDE using the Python package `sbi`, version 0.23.0. For comparison, we also trained a separate NDE using the hand-crafted MEA summary statistics extracted from the same simulations. Due to the complexity of the embedding network and the high dimensionality of the input, training and inference were approximately one hundred times slower than when using hand-crafted features. This, together with limitations in computational resources, prevented extensive architecture tuning.” (Supplementary Information, page 1)

We have mentioned our efforts with the embedding network, the negative results, and references to the new supplementary methods and Figure 5 in the discussion section:

“Yet, to assess whether data-driven summary statistics might improve inference, we also implemented a 2D-convolutional embedding network trained jointly with the NDE. However, this approach resulted in washed-out posteriors, poorer parameter recovery, and a dramatically increased computational cost of inference. Full details are provided in the Supplementary Methods and Supplementary Fig. 5.” (Lines 358-362).

We hope our additions are in line with the suggestion of the reviewer, and can help the SBI community.

We want to once again thank the reviewer for the tremendously insightful review and the motivating response to our rebuttal.